# OFFLINE REINFORCEMENT LEARNING FOR INTERVAL-CENSORED DATA

## ABSTRACT

This paper proposes a framework that applies reinforcement learning to multi-stage interval-censored data processing to develop an intelligent decision system capable of offering personalized behavioral recommendations based on observers' state and action variables. Interval-censored data is a common form of data encountered in practical data analysis, where observed results are only known to lie within certain intervals rather than exact values. This approach not only adapts to individual heterogeneity but also provides valuable decision support for personalized treatment. Experimental results demonstrate that this integrated approach effectively enhances individuals' longevity, providing a new method for personalized interventions and recommendations. This research is significant for the development of intelligent and personalized health management systems, offering valuable insights for future health sciences and intelligent decision systems.

## 1 INTRODUCTION

**Motivation.** With the advancement of medical research and clinical data analysis, interval-censored data has become an important component of survival analysis. However, existing reinforcement learning and policy optimization methods have mostly focused on right-censored data, neglecting the unique characteristics of interval-censored data. Interval-censored data is more complex than right-censored data because it involves greater uncertainty in the information. In practical medical research, particularly in cancer treatment and clinical follow-up studies, interval-censored data is commonly encountered. Effectively handling such data, especially when optimizing treatment strategies, remains a pressing challenge.

**Example 1.** In cancer treatment, due to periodic assessments (such as checking the status of cancer cells every month or every three months), doctors cannot precisely determine the exact time of cancer progression or transformation. They can only know the state of cancer between two successive assessments, making the survival time or disease progression time interval-censored rather than censored at a specific point. This interval-censored data is more complex to handle than right-censored data, as it involves uncertainty within a time range. It requires the integration of existing treatment information to optimize decision-making, ultimately aiming to improve patient survival rates and treatment outcomes more effectively.

**Example 2.** In clinical follow-up studies on hypertension, patients' blood pressure is typically measured once a week or every two weeks. However, the exact moment when a patient's blood pressure exceeds a certain threshold (i.e., the "conversion" event) cannot be accurately measured. Researchers can only know that the blood pressure exceeded the threshold between two measurements, but the exact moment of the increase remains undetermined. As a result, this conversion event constitutes interval-censored data.

**Challenges.** The application of reinforcement learning (RL) to interval-censored data presents several key challenges. These challenges arise primarily from the inherent complexities of interval-censored data and the limitations of existing methods designed for right-censored data. **Firstly,** interval-censored data introduces uncertainty as the event time is only known to lie within a specific time interval. This makes it difficult to directly apply RL methods, which typically rely on precise event times. To address this, we replace instantaneous rewards with the logarithm of the survival function to better capture this uncertainty. **Secondly,** individuals may have different observations at various stages, and some may exit or re-enter the study at different times. Incorporating this individual

heterogeneity and stage variability into the RL model is crucial for providing personalized treatment recommendations. **Finally,** optimizing decision strategies and ensuring that the value functions have desirable properties is a significant challenge. The value functions must account for the uncertainty in event times and converge to the true optimal strategy despite the noise in the data.

**Proposed Method.** This paper proposes an innovative reinforcement learning method to address optimal strategy estimation under interval-censored data structures. In contrast to traditional approaches that rely on instantaneous rewards, which are unavailable due to the nature of interval-censored data, we replace the instantaneous reward with the logarithm of the survival function. This adaptation allows the model to better account for the inherent characteristics of censored data. The method is applied to multi-stage decision problems, providing a robust solution to real-life challenges where such data issues arise. By leveraging this approach, the model can effectively optimize decision-making strategies across multiple stages, making it particularly suited for practical scenarios, such as clinical data analysis or other settings involving interval-censored data.

**Contribution.** The main contributions of this paper are as follows:
**(1) Development of a Reinforcement Learning Framework for Interval-Censored Data.** This paper introduces a novel reinforcement learning approach tailored to handle interval-censored data structures. By replacing instantaneous rewards with the logarithm of the survival function, the method accounts for the challenges posed by censored data, which are typically unaddressed in traditional approaches.
**(2) Application to Multi-Stage Decision Problems.** The proposed method is extended to multi-stage decision-making problems, providing a robust framework for optimizing strategies in real-world scenarios. This allows for dynamic and adaptive decision-making across various stages, making it highly applicable to fields such as clinical research, personalized healthcare, and other settings involving censored data.
**(3) Practical Impact on Personalized Health Management.** The framework's ability to effectively model personalized treatment decisions and optimize long-term outcomes is demonstrated, showing its potential for enhancing personalized interventions in medical settings.

## 2 RELATED WORK

**RL for Censored Survival Data.** Reinforcement learning (RL) has already been applied to survival data analysis, particularly in optimizing treatment strategies. These applications are especially relevant when dealing with censored survival data, which plays a critical role in personalized treatment decision-making. For example, Goldberg & Kosorok (2012) developed a Q-learning method for censored survival data to estimate optimal dynamic treatment regimes, and derived finite sample risk bounds associated with the generalization error of the estimated regime. Zhao et al. (2015) proposed a doubly robust estimator for expected survival time based on censored data, using outcome-weighted learning to estimate the optimal treatment regime. Jiang et al. (2017a) proposed two non-parametric estimators for the survival function of patients following a given treatment regime involving one or more decisions. Simoneau et al. (2019) extended the dynamic weighted ordinary least squares (dWOLS) approach to the censored data. Zhang et al. (2022) further extended DWSurv to estimate optimal DTRs of the censored data with multiple treatment options, which also inherits the shortcomings of DWSurv. Cho et al. (2023) developed a general dynamic treatment regime estimator for censored data, which allows the failure time to be conditionally independent of censoring and dependent on the treatment decision times.

**Gap in RL for Interval-Censored Data.** In the aforementioned literature, the focus is primarily on methods developed for right-censored data. However, in medical research and survival analysis, interval-censored data is often encountered, where the exact event time cannot be precisely observed, and only the time interval in which the event occurs is known Finkelstein (1986). For example, in clinical and medical follow-up studies, the failure time of interest is usually only known to lie between two examination times or within a certain time interval (Finkelstein, 1986; Sun, 2006). Compared to right-censored data, interval-censored data presents a more complex structure. While right-censored data typically includes some precisely observed failure times, interval-censored data introduces higher uncertainty in the information. This uncertainty poses significant challenges in identifying optimal treatment strategies in precision medicine. To the best of our knowledge, the application of RL to interval-censored data has not been extensively explored, particularly in the context of developing

individualized treatment strategies. Our research aims to fill this gap by developing a novel RL approach that can effectively handle interval-censored data. We propose a method that incorporates the inherent uncertainty in the event times and leverages the strengths of RL to identify optimal treatment policies. This innovative approach has the potential to significantly advance the field of personalized medicine by providing more accurate and tailored treatment recommendations.

## 3 NOTATION AND BACKGROUND

**Data Background.** Assume that at each stage, the response variable of interest is subject to interval censoring. Specifically, for the $i$-th individual (e.g., patient) at the $k$-th stage, the researchers can only observe that the event of interest occurs between two monitoring times, $U_{i,k}$ and $V_{i,k}$. More precisely, at time $U_{i,k}$, the event has not yet occurred, while at time $V_{i,k}$, the event is observed to have occurred, with the condition $U_{i,k} < V_{i,k}$. Due to the inability to conduct continuous real-time monitoring for each individual across multiple stages, the researchers are unable to pinpoint the exact time of the event's occurrence. Instead, they only know that it occurs within the interval $(U_{i,k}, V_{i,k}]$. The study involves $n$ individuals (patients), and the research problem is divided into $K$ distinct stages.

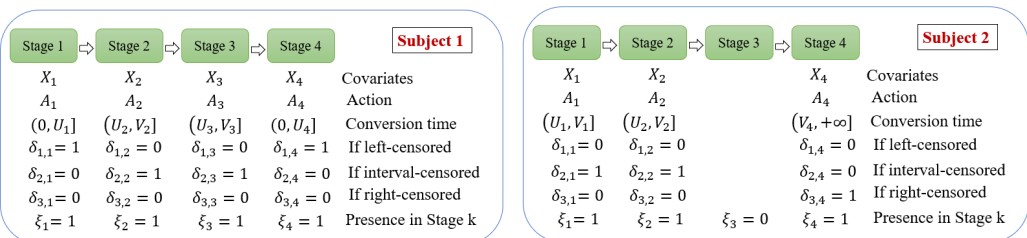

(a) Example 1 of multi-stage interval-censored data  (b) Example 2 of multi-stage interval-censored data

Figure 1: Examples of multi-stage interval-censored data

In particular, we consider situations in which individuals may enter the study at different stages, as well as situations in which they may leave the study for some reason. Therefore, we define the characteristic function $\xi_{i,k} = 1$ if the individual $i$ exists in the stage $k$ in the research experiment, otherwise $\xi_{i,k} = 0$. Specifically, there may be instances of stage skipping. For example, in cancer research, when the number of cancer cells does not significantly increase at the original site, but a phenomenon such as bone metastasis occurs, the patient may progress directly from stage 2 to stage 4. In such cases, $\xi_{i,3} = 0$. Moreover, for individuals whose first observation upon entering stage $k$ reveals that the event has already occurred, the event time is within the interval $[0, U]$, a scenario referred to as left censoring. This corresponds to the indicator function $\delta_{1,i,k} = 1, \delta_{2,i,k} = 0, \delta_{3,i,k} = 0$. When the event time in stage $k$ falls within the interval $(U, V]$, it is referred to as interval censoring, with the corresponding indicator values $\delta_{1,i,k} = 0, \delta_{2,i,k} = 1, \delta_{3,i,k} = 0$. In particular, if a patient exits the study for any reason (excluding death) during stage $k$, and the detection at time $V$ in stage $k$ shows that the event has not yet occurred, the event time is considered to be within the interval $(V, \infty)$, which is known as right censoring. In this case, the indicator function is $\delta_{1,i,k} = 0, \delta_{2,i,k} = 0, \delta_{3,i,k} = 1$. Additionally, for individuals in stage $k$ with $\xi_{i,k} = 1$, it is always the case that $\delta_{1,i,k} + \delta_{2,i,k} + \delta_{3,i,k} = 1$, ensuring that each individual's data at each stage is categorized into one of the three types of censoring: left, interval, or right censoring.

**Multi-Stage Interval-Censored Data.** Let's first briefly review the basic concepts and notation of reinforcement learning in the context of complete data. Let $\mathbf{X}_{0,t} \in \mathbb{X}$ be the time-varying covariates collected at time point $t$, $A_{0,t} \in \mathcal{A}$ denote the action taken at time $t$, and $Y_{0,t}$ stand for the immediate reward observed. Here, $\mathbb{X}$ and $\mathcal{A}$ denote the state and action space respectively. We assume $\mathbb{X}$ is a subspace of $\mathbb{X}^p$ where $p$ is the number of state vectors and $\mathcal{A}$ is a discrete space $\{0, 1, \ldots, m-1\}$ where $m$ denotes the number of actions. Suppose the system satisfies the following Markov Assumption (MA), $Pr(\mathbf{X}_{0,t+1}|\mathbf{X}_{0,t} = \mathbf{x}, A_{0,t} = a, \{\mathbf{X}_{0,j}, A_{0,j}\}_{0 \leq j < t}) = P(\mathbf{X}_{0,t+1}|\mathbf{x}, a)$, for some transition function $P$. That is, the above formula is the conditional probability of the next state given the condition of the current state-action pair.

Assuming that a total of $K$ stages of study are conducted, then individual trajectories can be expressed as $\mathcal{D} = \{\mathbf{X}_{0,k}, A_{0,k}, U_{0,k}, V_{0,k}, \delta_{1,0,k}, \delta_{2,0,k}, \delta_{3,0,k}, \xi_{0,k}\}_{k=1}^{K}$. Note that the dataset $\mathcal{D}$ for the individual includes observations collected over $K$ time points, with actions selected according to a behavior policy. Define the conditional survival function for failure time $Y$ given covariates (state and action) as $S_Y(y|\boldsymbol{x}, a) = P(Y \geq y|\mathbf{X} = \boldsymbol{x}, A = a)$, which is the probability that $Y \geq y$ given $\mathbf{X}, A$.

# 4 EVALUATION FOR INTERVAL-CENSORED DATA

**Summary.** This section discusses the development of a reinforcement learning framework based on the MDP assumption for multi-stage interval-censored data, aiming to find the optimal decision policy. Additionally, to effectively estimate the immediate reward in the context of interval-censored data, a Sieve-based maximum likelihood estimation method is proposed for constructing the reward estimates.

## 4.1 METHODOLOG

We assume that the dataset, $\mathcal{D}_n$, consists of $n$ trajectories, $\mathcal{D}_n = \{\mathcal{D}^i\}_{i=1}^{n}$, with $\mathcal{D}^i = \{\mathbf{X}_{i,k}, A_{i,k}, U_{i,k}, V_{i,k}, \delta_{1,i,k}, \delta_{2,i,k}, \delta_{3,i,k}, \xi_{i,k}\}_{k=1}^{K}$. Each trajectory, $\mathcal{D}^i$, is an i.i.d. copy of $\mathcal{D}$ described in Section 3. Recall that $\{A_{i,t}\}_{t=1}^{K}$, the actions in $\mathcal{D}$, are selected by the behavior policy, $\pi$. In the following, the expectation, $\mathbb{E}$, without the subscript is with respect to the distribution of the trajectory, $\mathcal{D}$, under the behavior policy. The transition probability, denoted by $P$, is defined for any measurable set $\mathcal{B} \in \mathbb{X}$ as $P(\mathcal{B} \mid \mathbf{x}, a) = \Pr(\mathbf{X}_{0,t+1} \in \mathcal{B} \mid \mathbf{X}_{0,t} = \mathbf{x}, A_{0,t} = a)$ where $\mathbf{X}_{0,t}$ and $A_{0,t}$ represent the state and action at time $t$, respectively. This transition probability is time-invariant, in accordance with the Markov assumption (MA). Let $p(\mathbf{x}' \mid \mathbf{x}, a)$ denote the transition density with respect to a chosen reference measure (e.g., the counting measure when $\mathbb{X}$ is discrete).

**The Reward in Interval-Censored Data.** To evaluate cumulative rewards under a pre-specified time-invariant Markovian policy $\pi$, we need to define an appropriate immediate reward. If the rewards were precisely observed, these exact values could naturally serve as the reward. However, due to the interval-censored nature of the data, such precise observations are unavailable. To address this, we define the immediate reward as the logarithm of the conditional probability $Y_{0,k} \geq T_k$ given $\mathbf{X}_{0,k}, A_{0,k}$, i.e., $\log\left(S_{Y_{0,k}}(T_k|\mathbf{X}_{0,k}, A_{0,k})\right) = -\Lambda_{Y_{0,k}}(T_k|\mathbf{X}_{0,k}, A_{0,k})$, where $\Lambda_{Y_{0,k}}$ denotes the cumulative hazard function of $Y_{0,k}$, and $T_k$ represents the ideal duration for which an individual should remain in stage $k$ without transitioning to the next stage. Since the survival function takes values in the range $[0, 1]$, it is evident that the immediate reward defined above is bounded. The choice of the logarithmic survival function as the immediate reward is not only a practical substitution but also offers several advantages. It effectively quantifies the likelihood of achieving a threshold survival time, aligning with the study's objective of maximizing survival time. Additionally, the survival function and its logarithm are widely used in survival analysis, making them mathematically tractable for policy optimization.

**The Action-Value Function in Interval-Censored Data.** Using the definition of the above reward function, the cumulative discounted reward beyond stage $t$ may be written as $-\sum_{k=t}^{K} \gamma^{k-t} \Lambda_{Y_{0,k}}(T_k|\mathbf{X}_{0,k}, A_{0,k})$. The goal is to identify an optimal policy that maximizes the probability of the survival time exceeding a specified threshold. With this definition of immediate reward, the action-value function (Q-function) $Q^\pi(\mathbf{x}, a)$ under policy $\pi(\boldsymbol{x})$ for $\boldsymbol{x} \in \mathbb{X}$, is expressed as the expected discounted cumulative reward:

$$-\mathbb{E}^\pi \left[ \sum_{k=t}^{K} \gamma^{k-t} \Lambda_{Y_{0,k}}(T_k|\mathbf{X}_{0,k}, A_{0,k}) \Big| \mathbf{X}_{0,t} = \boldsymbol{x}, A_{0,t} = a \right]$$

where $\gamma$ is the discount factor. This is the expected cumulative discounted reward if taking treatment $a$ at state $\boldsymbol{x}$ at stage $t$ and then following $\pi(\boldsymbol{x})$ until the end of the study.

**The Bellman Equation.** Let us denote immediate reward $-\Lambda_{Y_{0,k}}(T_k|\mathbf{X}_{0,k}, A_{0,k}) = R_k$. For $\gamma < 1$, the optimal action-value function is defined as $Q^*(\boldsymbol{x}, a) = \max_\pi Q^\pi(\boldsymbol{x}, a)$, and satisfies the

following recursive relationship:

$$Q^*(\mathbf{x}, a) = E\left\{ R_{t+1} + \gamma \max_{a' \in \mathcal{A}} Q^* \left(\mathbf{X}_{t+1}, a'\right) \mid \mathbf{X}_t = \mathbf{x}, A_t = a \right\}, \tag{1}$$

where $\boldsymbol{x} \in \mathbb{X}$, and $a \in \mathcal{A}$. The function $Q^*(\boldsymbol{x}, a)$ is influenced by the transition probability distribution, specifically the probability of reaching state $c$ given that $(\mathbf{X}_t, A_t) = (\boldsymbol{x}, a)$. A policy $\pi^*(\boldsymbol{x})$, $\boldsymbol{x} \in \mathbb{X}$, is called an optimal dynamic treatment regime if it satisfies $Q^*(\mathbf{x}, a) = Q^{\pi^*}(\mathbf{x}, a)$. For a given $\boldsymbol{x} \in \mathbb{X}$, he optimal policy can equivalently be determined as $\pi^*(\boldsymbol{x}) = \arg\max_{a \in \mathcal{A}} Q^*(\boldsymbol{x}, a)$.

Equation (1) is referred to as the Bellman equation for $Q^*(\boldsymbol{x}, a)$ (Sutton & Barto, 1999; Si et al., 2004). The discount factor $\gamma$ determines the trade-off between immediate and long-term effects of treatments on the action-value function. When $\gamma = 0$, the focus is solely on maximizing the immediate reward, disregarding any impact the action may have on future rewards or outcomes. As $\gamma \to 1$, greater weight is placed on future rewards.

### 4.2 ESTIMATION OF THE SURVIVAL FUNCTION

**The Maximum Likelihood Function.** First, we can solve the survival function estimation of each stage by maximum likelihood estimation. For this part of processing, we can first construct the likelihood function corresponding to stage $k$ as follows:

$$L_k = \prod_{i=1}^n \left\{ \left(1 - S(U_{i,k}|\mathbf{X}_{i,k}, A_{i,k})\right)^{\xi_{i,k}\delta_{1,i,k}} \left(S(U_{i,k}|\mathbf{X}_{i,k}, A_{i,k}) - S(V_{i,k}|\mathbf{X}_{i,k}, A_{i,k})\right)^{\xi_{i,k}\delta_{2,i,k}} \right.$$

$$\left. \left(S(V_{i,k}|\mathbf{X}_{i,k}, A_{i,k})\right)^{\xi_{i,k}\delta_{3,i,k}} \right\},$$

and the logarithmic likelihood function is expressed as $\mathcal{L}_k$:

$$\sum_{i=1}^n \left\{ \xi_{i,k}\delta_{1,i,k} \log\left(1 - S(U_{i,k}|\mathbf{X}_{i,k}, A_{i,k})\right) + \xi_{i,k}\delta_{2,i,k} \log\left(S(U_{i,k}|\mathbf{X}_{i,k}, A_{i,k}) - S(V_{i,k}|\mathbf{X}_{i,k}, A_{i,k})\right) \right.$$

$$\left. + \xi_{i,k}\delta_{3,i,k} \log\left(S(V_{i,k}|\mathbf{X}_{i,k}, A_{i,k})\right) \right\} = \sum_{i=1}^n \xi_{i,k} \log\left(S(Y_{i,k}^L|\mathbf{X}_{i,k}, A_{i,k}) - S(Y_{i,k}^R|\mathbf{X}_{i,k}, A_{i,k})\right),$$

where $Y_{i,k}^L = \delta_{1,i,k} \times 0 + \delta_{2,i,k} \times U_{i,k} + \delta_{3,i,k} \times V_{i,k}$, and $Y_{i,k}^R = \delta_{1,i,k} \times U_{i,k} + \delta_{2,i,k} \times (V_{i,k}) + \delta_{3,i,k} \times \infty$.

**Bernstein polynomial parameterization.** Here we considers the use of Bernstein polynomials to approximate the baseline cumulative hazard function $\Lambda_0(\cdot)$ in survival analysis, as discussed below by following Zhou et al. (2017). The corresponding method for estimating the survival function is referred to as the sieve method. The sieve method, which approximates the survival function through successive refinement, is effective for estimating survival functions in most survival data models. Define the parametric space of $\boldsymbol{\vartheta}$ to be $\Theta = \{\boldsymbol{\vartheta} = (\boldsymbol{\eta}, \Lambda_0) \in \mathcal{B} \otimes \mathcal{M}\}$, where $\mathcal{B} = \{\boldsymbol{\eta}|\boldsymbol{\eta} \in R^{2p+1}, \|\boldsymbol{\eta}\| \leq M\}$ with $M$ is a positive constant and $\mathcal{M}$ is the collection of all bounded and continuous nondecreasing, nonnegative functions over the interval $[u, v]$ with $0 \leq u < v < \infty$. In general, the values of $u$ and $v$ are the minimum and maximum values of the observed data. Also define the sieve space $\Theta_n = \{\boldsymbol{\vartheta}_n = (\boldsymbol{\eta}, \Lambda_n) \in \mathcal{B} \otimes \mathcal{M}_n\}$, where

$$\mathcal{M}_n = \left\{ \Lambda_n = \sum_{k=0}^m \phi_k^* B_k(t, m, u, v) : \sum_{0 \leq k \leq m} |\phi_k^*| \leq M_n, 0 \leq \phi_0^* \leq \phi_1^* \leq \ldots \leq \phi_m^* \right\}$$

with the $\phi_k^*$'s being some parameters, $M_n = O(n^a)$ for some $a > 0$ controlling the size of $\Theta_n$, and the Bernstein basis polynomials of degree $m = o(n^s)$ for some $s \in (0, 1)$ is

$$B_k(t, m, u, v) = \binom{m}{k} \left(\frac{t-v}{u-v}\right)^k \left(1 - \frac{t-v}{u-v}\right)^{m-k},$$

with $k = 0, \ldots, m$. Note that due to the nonnegativity and monotonicity features of $\Lambda_0(\cdot)$, we need the constraint $0 \leq \phi_0^* \leq \phi_1^* \leq \ldots \leq \phi_m^*$, but it can be easily attained by the reparameterization $\phi_0^* = e^{\phi_0}$ and $\phi_k^* = \sum_{i=0}^k e^{\phi_i}, 1 \leq i \leq m$.

## 5 THEORETICAL RESULTS

**Summary.** In this section, we examine the theoretical properties of action-value function estimation in the context of interval-censored data. We begin by presenting the essential regularization conditions that form the foundation of our analysis.

**Assumption 5.1.** For each stage $k$, there exists a constant $\eta > 0$ such that $Pr(V_k - U_k \geq \eta) = 1$. Additionally, the union of the supports of $U_k$ and $V_k$ is contained in the interval $[\sigma, \tau]$, where $0 < \sigma < \tau < +\infty$.

**Assumption 5.2.** The distribution of $\mathbf{X}$ has bounded support and is not concentrated on any proper subspace of $\mathbb{X}$. Also, $E\{\text{var}(\mathbf{X} \mid U)\}$ and $E\{\text{var}(\mathbf{X} \mid V)\}$ are positive definite;

**Assumption 5.3.** For $r = 1$ or $2$, the function $\Lambda_0 \in \mathcal{M}$ is continuously differentiable up to order $r$ in $[\sigma, \tau]$, with the first derivative being strictly positive, and satisfies $\alpha^{-1} < \Lambda_0(\sigma) < \Lambda_0(\tau) < \alpha$ for some positive constant $\alpha$.

**Assumption 5.4.** The sample trajectories $\left\{\mathcal{D}^i\right\}_{i=1}^n$ are generated from (possibly $n$ different) Markov decision processes satisfying
(1) (**Markovianity**) For any $i \in \{1, \ldots, n\}$ and $k \in \{1, \ldots, K\}$, it holds that $\mathbf{X}_{i,k}$ is independent of $\{\mathcal{H}_j\}_{0 \leq s \leq k-1}$, conditional on $(\mathbf{X}_{i,k-1}, A_{i,k-1})$, where $\{\mathcal{H}_j\}$ denots $\{(\mathbf{X}_{i,s}, A_{i,s}, U_{i,s}, V_{i,s}, \delta_{1,i,s}, \delta_{2,i,s}, \delta_{3,i,s}, \xi_{i,s})\}$.
(2) (**Time-Homogeneity**) The conditional density $\Pr(\mathbf{X}_{i,k+1} \mid \mathbf{X}_{i,k}, A_{i,t})$ is the same over $k$ for any $i \in \{1, \ldots, n\}$.
(3) (**Conditional Mean Independence(CMI)**) For any $i \in \{1, \ldots, n\}$ and $k \in \{1, \ldots, K\}$, it holds that

$$\mathbb{E}\left[R_{i,k} \mid \mathbf{X}_{i,k} = \boldsymbol{x}, A_{i,k} = a, \{\mathcal{H}_s\}_{0 \leq s < k}\right] = \mathbb{E}\left[R_{i,k} \mid \mathbf{X}_{i,k} = \boldsymbol{x}, A_{i,k} = a\right] = r_i(\boldsymbol{x}, a),$$

where $R_{i,k} = \log S_{Y_{0,k}}(t \mid \mathbf{X}_{i,k}, A_{i,k})$, for some bounded reward function $r_i(\boldsymbol{x}, a)$.

Assumptions 5.1-5.3 are widely adopted in the study of interval-censored data (Huang & Rossini, 1997; Zhang et al., 2010) and are generally satisfied in practical scenarios. Assumptions 5.4 is commonly seen in the reinforcement learning framework, as demonstrated in the works of Chen et al. (2022); Shi et al. (2022). Estimating $S_{Y_{0,k}}(\cdot|\widetilde{\mathbf{X}}_k, \widetilde{A}_k)$ can be challenging because there are at least two potential treatments at each stage, and $Y_{0,k}$ can not be observed accurately due to interval censoring. Here, we can adopt various estimation methods, including but not limited to the traditional Cox proportional hazards model, accelerated failure time model, and neural network-based approaches. We summarize the asymptotic normality property of the proposed Q-function based on the estimation error in the following Theorem.

**Theorem 5.5.** *Under the Assumption 5.1-5.4, we denote $\hat{\Lambda}_{T_k}(t|\tilde{\mathbf{X}}_k, \tilde{A}_k) - \Lambda_{T_k}(t|\tilde{\mathbf{X}}_k, \tilde{A}_k) = \zeta_k(\mathcal{D}_n)$ and $\hat{Q}^\pi(\tilde{\mathbf{x}}_t, \tilde{a}_t) - Q^\pi(\mathbf{x}_t, a_t) = \delta_t^\pi$. If $\zeta_k(\mathcal{D}_n)$ are mutually independent and satisfy $\mathbb{E}[\zeta_k(\mathcal{D}_n)] = 0$ and $\mathbb{E}[\zeta_k^2(\mathcal{D}_n)] < +\infty$, $\forall k, \mathcal{D}_n$, then $\forall \pi \in \Pi$ we have*

$$\mathbb{E}\left[\delta_t^\pi\right] = 0, \text{ and } Var\left[\delta_t^\pi\right] \leq \sum_{k=t}^{\bar{K}} \gamma^{2k-2t} \sup_k \mathbb{E}[\zeta_k^2(\mathcal{D}_n)]. \tag{2}$$

**Remark.** This theorem establishes that under the given assumptions, the estimator $\widehat{Q}^{\pi^*}(\widetilde{\mathbf{x}}, \widetilde{a})$ is unbiased, with its expected value equal to the true action-value function. Additionally, it quantifies the variance of the estimator, which depends on the underlying distribution, the optimal policy, and the discount factor. The results provide insight into the reliability and behavior of the estimator, with the variance influenced by the temporal dynamics and transition probabilities in the model.

## 6 A SIMULATION STUDY

In this paper, the data structure observed by the researchers should be $(U_{i,k}, V_{i,k}, \delta_{1,i,k}, \delta_{2,i,k}, \delta_{3,i,k}, \xi_{i,k}, \mathbf{X}_{i,k})$, the true response variable $Y_{i,k}$ located between $Y_{i,k}^L = \delta_{1,i,k} \times 0 + \delta_{2,i,k} \times U_{i,k} + \delta_{3,i,k} \times V_{i,k}$ and $Y_{i,k}^R = \delta_{1,i,k} \times U_{i,k} + \delta_{2,i,k} \times (V_{i,k}) + \delta_{3,i,k} \times \eta$, where $\eta$ is a sufficiently large constant that we take the value of $10^6$ here. In the simulation study,

we consider that there are $K = 10$ stages. For the initial phase (i.e. $k = 1$), we generate the two-dimensional covariates $X_{i,1}$ from the standard normal distribution $N(\mathbf{0}, \boldsymbol{I}_2)$. For each phase, you can choose an action from 0 or 1. The iterative formula for $k = 1, 2, \ldots, 9$, $\mathbf{X}_{i,k+1}$ is as follows:

$$\mathbf{X}_{i,k+1} = \begin{bmatrix} \frac{3}{4}(2\mathbf{A}_{i,k} - 1) & 0 \\ 0 & \frac{3}{4}(2\mathbf{A}_{i,k} - 1) \end{bmatrix} \mathbf{X}_{i,k} + \boldsymbol{\epsilon}_{i,k},$$

where $\epsilon_{i,k}$ is the random error term and its distribution satisfies the multivariate normal distribution $N(\mathbf{0}, 1/4\boldsymbol{I}_2)$.

**Cox Proportional Hazards model.** Assume that the response variable $Y_{i,k}$ follows the follow Cox Proportional Hazards model $\Lambda(Y_{i,k}|\mathbf{X}_{i,k}, A_{i,k}) = \Lambda_0(t)\exp(\mathbf{Z}_{i,k}^\top\boldsymbol{\eta})$, where $\mathbf{Z}_{i,k} = (\mathbf{X}_{i,k}^\top, A_{i,k}, A_{i,k}\mathbf{X}_{i,k}^\top)^\top$, the corresponding parameter set as $\eta = (-2, 1, -0.5, 4, -2)^\top$, and the cumulative hazard function set as $\Lambda_0(t) = t$.

**Accelerated Failure Time Model.** Assume that the response variable $Y_{i,k}$ was generated under the transformation model

$$\log Y_{i,k} = -(\mathbf{Z}_{i,k}^\top\boldsymbol{\eta}) + \epsilon_{i,k}, \tag{3}$$

where $\mathbf{Z}_{i,k} = (\mathbf{X}_{i,k}^\top, A_{i,k}, A_{i,k}\mathbf{X}_{i,k}^\top)^\top$, the corresponding parameter set as $\eta = (-2, 1, -0.5, 4, -2)^\top$, and the error $\epsilon_{0,k}$ follows the standard normal distribution.

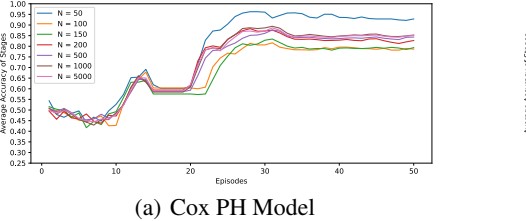 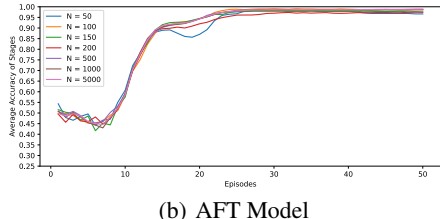

(a) Cox PH Model         (b) AFT Model

Figure 2: Average Accuracy of each Stage to Episodes

To simulate the time boundary for each stage, we impose an upper bound of 4 on the generated $Y_{i,k}$, meaning the generated $Y_{i,k}$ will be truncated at 4. Subsequently, we generate $J_{i,k}$ following a uniform distribution $U(0.5, 1.5)$ and $H_{i,k}$ following a uniform distribution $U(3, 4.5 - 0.1k)$. This setup is intended to simulate right-censoring caused by an increasing number of people choosing not to wear the device as the order increases. We then define:

- When $Y_{i,k} \leq J_{i,k}$: $\delta_{1,i,k} = 1$ and $U_{i,k} = J_{i,k}$.

- When $J_{i,k} \leq Y_{i,k} \leq J_{i,k} + H_{i,k}$: $\delta_{2,i,k} = 1$ and $U_{i,k} = J_{i,k}$, $V_{i,k} = J_{i,k}$.

- When $Y_{i,k} \geq J_{i,k} + H_{i,k}$: $\delta_{3,i,k} = 1$ and $U_{i,k} = J_{i,k} + H_{i,k}$.

We set $\alpha = \frac{1}{2}$. For DDPG-based neural networks, we use a reinforcement learning agent with a batch size of 32, a replay buffer of 6400, a learning rate of 0.001, a soft update $\tau$ of 0.01, and a discount factor $\gamma$ of 0.99. We use the DDPG algorithm, with the detailed steps provided in Algorithm 1 in the appendix A.2. The network is updated at every step and torrents are used to ensure repeatability. We did 100 episodes in each condition. The results of our analysis are presented as follows:

As shown in Figure 2, it can be observed that the accuracy of the formulated strategies increases with the number of training iterations. Moreover, the AFT model demonstrates higher accuracy, and as the sample size increases, the accuracy stabilizes and improves. From Figure 3, we can see that the rewards obtained also increase with training, corresponding to the improvement in accuracy. In Figure 6, the change in the right-censoring rate in our simulation is illustrated. It is evident that the censoring rates under the two models are close to each other. Furthermore, as the number of training iterations increases and the formulated strategies stabilize, the right-censoring rate also stabilizes. Additionally, with larger sample sizes, the censoring rate becomes more stable. Next, we will present the results at different sample sizes in phases.

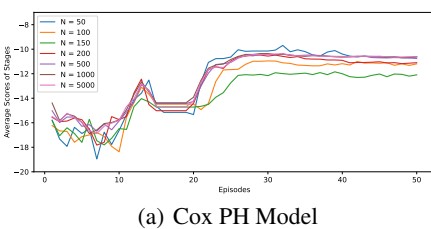 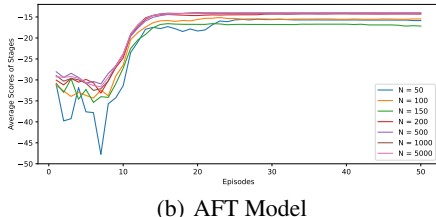

(a) Cox PH Model                        (b) AFT Model

Figure 3: Total Scores of each Stage to Episodes, which demonstrates that our method converges as the number of training episodes increases.

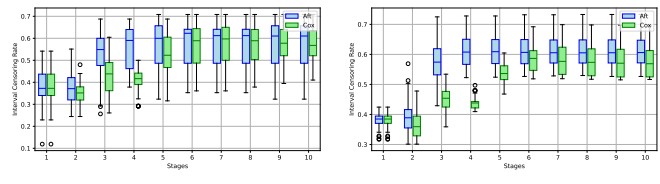 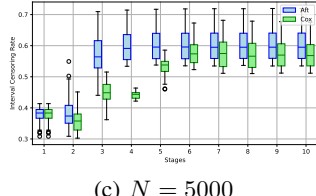

(a) $N = 50$              (b) $N = 1000$              (c) $N = 5000$

Figure 4: Interval censoring rates of all episodes for each stage under different $N$ using Aft and Cox Model

In Figure 4, we can observe that whether using the Cox PH Model or the AFT Model, the interval censoring rate at each stage tends to stabilize as the sample size increases, and the ratios under the fitting of the two models are similar. In Figure 7, we can see that regardless of whether the Cox PH Model or the AFT Model is used, the left censoring rate at each stage tends to stabilize as the sample size increases, and the ratios under the fitting of the two models are similar. In our simulated scenario, the initial stage shows a higher interval censoring rate, approximately 65%, which later decreases to around 40%. As shown in Figure 8 and Figure 9, the accuracy of decision-making tends to stabilize as the training sample size increases. The accuracy when using the Cox PH Model is consistently above 75%, and when using the AFT Model, it is consistently above 90%. Moreover, when the sample size reaches 5000, the accuracy for both models exceeds 95%. This indicates that under this simulation, the fitting effect of the AFT Model is relatively better, as it more accurately reflects the relationship between the reward values, allowing for the formulation of correct strategies. Based on the experimental results, our proposed method demonstrates excellent performance in learning the specified strategy for both large and small sample sizes, achieving high accuracy. We also provide the left censoring rate and a comparison of the accuracy of the two methods at each stage in a boxplot in Appendix A.4.

## 7 AN APPLICATION

In this section, we apply the framework proposed in earlier sections to analyze survival time data from breast cancer patients in the Surveillance, Epidemiology, and End Results (SEER) program, a widely recognized resource in cancer research (https://seer.cancer.gov). The SEER program was established as one of the first steps in the War on Cancer declared by President Nixon's Administration and began collecting information on January 1, 1973 in some of US states with other areas added to the SEER database over the years. After the year 2000, the SEER captured approximately 25% of all cancers diagnosed in the United States each year.

The data considered in the following analysis were submitted in November 2020 and released in April 2021. The dataset includes information on 95,056 breast cancer patients diagnosed between 2010 and 2015, along with 11 covariates, including sex, year of diagnosis, race, primary site, tumor size, breast T stage, and others. For the analysis here, we are mainly interested in the survival time of this stage, the failure time of interest, of the cancer patient, on which only interval-censored data are available due to the periodic collection nature of the data. In this study, breast cancer was classified

into stages based on immunohistochemical results. Specifically, the following stages were identified: Stage 1, HR+/HER2+ (Luminal B); Stage 2, HR+/HER2- (Luminal A); Stage 3, HR-/HER2+ (HER2 enriched); Stage 4, HR-/HER2- (Triple Negative). In addition, the focus of this study is on the action of radiation. Based on the official documentation, we define $X_{i,k}$ as the information of the $i$-th individual at the $k$-th stage, $A_{i,k}$ as a binary variable indicating treatment (1 for treated, 0 for not treated), $U_{i,k}$ as the left endpoint of the observed event interval, and $V_{i,k}$ as the right endpoint of the observed event interval.

We applied the Cox proportional hazards model to estimate the survival function and subsequently used reinforcement learning methods, including Deep Q-Network (DQN) (Mnih et al., 2015), Discrete Conservative Q-Learning (DiscreteCQL) (Kumar et al., 2020), Discrete Batch-Constrained Q-Learning (DiscreteBCQ) (Fujimoto et al., 2019), and Double Deep Q-Network (DoubleDQN) Van Hasselt et al. (2016), through the d3rlpy library for fitting.

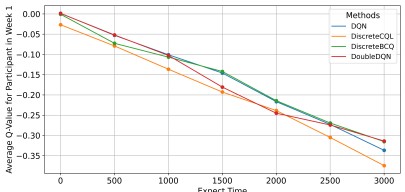

Figure 5: Trajectories of the average Q-value in week 1 over expect time based on different methods.

Table 1: Significant Variables with p-value < 0.005 in Each Stage

| Variable | Stage 1 | Stage 2 | Stage 3 | Stage 4 |
|---|---|---|---|---|
| X1 | 1.53e-29 | | 3.06e-04 | |
| X10 | 0 | 2.04e-29 | 4.73e-40 | 1.10e-81 |
| X11 | 3.61e-26 | | 2.70e-03 | |
| X2 | 1.52e-12 | | 7.37e-06 | 3.23e-07 |
| X6 | 0 | 1.11e-14 | 2.24e-20 | 1.11e-33 |

X1 to X11 represent the following variables: X1 indicates race (e.g., White, Black, Asian, etc.), X2 represents the original ethnicity classification (e.g., Hispanic and Non-Hispanic), X6 refers to tumor size and external evaluation, and X10 and X11 pertain to clinical features and staging of breast cancer. According to the results shown in Table 1, X1 is significant in stages 1 and 3, X10 is significant across all four stages, X11 is significant in stages 1 and 3, X2 is significant in stages 1, 3, and 4, and X6 is significant in all stages. Overall, X10 and X6 are significant in all stages, suggesting that their influence is consistent and important throughout the different stages. The significance of X2, X1, and X11 appears only in certain stages, indicating that their impact may vary across stages. The specific code can be referenced in our supplementary material. In summary, our method provides a probabilistic analysis of interval-censored data under reinforcement learning and presents the significant features at each stage.

## 8 CONCLUSION

This paper presents a novel reinforcement learning approach for handling multi-stage interval-censored data. In such data structures, exact reward signals are unavailable, requiring indirect methods for decision-making optimization. To address this, we define the immediate reward as the logarithmic survival function, transforming incomplete reward information into a usable form within survival analysis. Additionally, we replace the traditional survival function with a sieve estimate to better handle the challenges of interval censoring, improving the model's robustness and predictive capability. Building on this, we construct a Q-function tailored to this framework, optimizing treatment policies and identifying the best strategies across multiple decision stages. Overall, the proposed method provides a new solution for dealing with uncertainty in interval-censored data, combining reinforcement learning with survival analysis to enhance decision-making in medical research. This approach offers promising potential for improving treatment strategies and precision in clinical decision-making.

However, we are cognizant of the limitations inherent in our study. The model's effectiveness may be subject to the specificities of the data and may require further validation across various cancer types and patient populations. Future work should explore the integration of this system with emerging technologies such as digital twins to push the boundaries of precision medicine in oncology.

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

# A APPENDIX

## A.1 TWO EXAMPLE MODELS FOR SIEVE ESTIMATION

Here, We give two common examples, namely the Cox proportional hazards model and the Accelerated Failure Time model.

**1. Cox proportional hazards model**

Suppose that the reward at time stage $k$ satisfies the Cox PH model of the current state $\mathbf{X}_{0,k}$ and the current processing $A_{0,k}$. In this section, we will assume that given $\mathbf{X}_{0,k}$ and $A_{0,k}$, $Y_{0,k}$ follows the proportional hazards model or its cumulative hazard function has the form

$$\Lambda(Y_{0,k}|\mathbf{X}_{0,k}, A_{0,k}) = \Lambda_0(Y_{0,k}) \exp(\mathbf{Z}_{0,k}^\top \boldsymbol{\eta}),$$

where $\Lambda_0$ is the cumulative baseline hazard function, $\mathbf{Z}_{0,k} = (\mathbf{X}_{0,k}^\top, A_{0,k}, A_{0,k}\mathbf{X}_{0,k}^\top)^\top$ as in Jiang et al. (2017b) and $\boldsymbol{\eta} = (\boldsymbol{\beta}_1^\top, \beta_2, \boldsymbol{\beta}_3^\top)^\top$.

With the use of the Bernstein polynomials method, the log-likelihood function can be rewritten as

$$\mathcal{L}_k(\boldsymbol{\eta}, \boldsymbol{\phi}) = \sum_{i=1}^n \xi_{i,k} \log \left( \exp \left( -\Lambda_n(Y_{i,k}^L) \exp(\mathbf{Z}_{0,k}^\top \boldsymbol{\eta}) \right) - \exp \left( -\Lambda_n(Y_{i,k}^R) \exp(\mathbf{Z}_{0,k}^\top \boldsymbol{\eta}) \right) \right),$$

where $\boldsymbol{\phi} = (\phi_0, \ldots, \phi_m)^\top$. Thus,

$$(\widehat{\boldsymbol{\eta}}, \widehat{\boldsymbol{\phi}}) = \arg \max_{\boldsymbol{\eta}, \boldsymbol{\phi}} \mathcal{L}_k(\boldsymbol{\eta}, \boldsymbol{\phi}),$$

and

$$\widehat{\Lambda}(y|\mathbf{X}_{i,k}, A_{i,k}) = \widehat{\Lambda}_n(y) \exp(\mathbf{Z}_{0,k}^\top \widehat{\boldsymbol{\eta}}).$$

**2. Accelerated Failure Time Model**

Suppose that the reward at time stage $k$ satisfies the Accelerated Failure Time (AFT) model of the current state $\mathbf{X}_{0,k}$ and the current processing $A_{0,k}$. In this section, we will assume that

$$\log Y_{0,k} = -(\mathbf{Z}_{0,k}^\top \boldsymbol{\eta}) + \epsilon_{0,k}, \tag{4}$$

where $\mathbf{Z}_{0,k} = (\mathbf{X}_{0,k}^\top, A_{0,k}, A_{0,k}\mathbf{X}_{0,k}^\top)^\top$, $\boldsymbol{\eta} = (\boldsymbol{\beta}_1^\top, \beta_2, \boldsymbol{\beta}_3^\top)^\top$ and $\epsilon_{0,k} \overset{i.i.d.}{\sim} f$ are the error, such models are also sometimes referred to as log-linear models. The equivalent hazard-based specification of the AFT model is

$$\Lambda(Y_{0,k}|\mathbf{X}_{0,k}, A_{0,k}) = \Lambda_0(\exp(\mathbf{Z}_{0,k}^\top \boldsymbol{\eta})Y_{0,k}) \exp(\mathbf{Z}_{0,k}^\top \boldsymbol{\eta})$$

where $\Lambda_0(\cdot)$ is the baseline hazard function corresponding to $\mathbf{X}_{0,k} = \mathbf{0}$, $A_{0,k} = 0$, and $(\boldsymbol{\beta}_1^\top, \beta_2, \boldsymbol{\beta}_3^\top)$ is the same log(time ratio) vector as in equation 4 (Kalbfleisch & Prentice, 2011).

Similarly, the Bernstain polynomial can be used to parameterize the baseline cumulative hazard function, and the log-likelihood function $\mathcal{L}_k(\boldsymbol{\eta}, \boldsymbol{\phi})$ can be rewritten as

$$\sum_{i=1}^{n} \xi_{i,k} \log \Big( \exp \big( - \Lambda_0(\exp(\mathbf{Z}_{0,k}^{\top}\boldsymbol{\eta})Y_{i,k}^L) \exp(\mathbf{Z}_{0,k}^{\top}\boldsymbol{\eta}) \big) - \exp \big( - \Lambda_0(\exp(\mathbf{Z}_{0,k}^{\top}\boldsymbol{\eta})Y_{i,k}^R) \exp(\mathbf{Z}_{0,k}^{\top}\boldsymbol{\eta}) \big) \Big),$$

where $\boldsymbol{\phi} = (\phi_0, \ldots, \phi_m)^{\top}$. Thus,

$$(\widehat{\boldsymbol{\eta}}, \widehat{\boldsymbol{\phi}}) = \arg \max_{\boldsymbol{\eta}, \boldsymbol{\phi}} \mathcal{L}_k(\boldsymbol{\eta}, \boldsymbol{\phi}),$$

and

$$\widehat{\Lambda}(y|\mathbf{X}_{i,k}, A_{i,k}) = \widehat{\Lambda}_n(\exp(\mathbf{Z}_{0,k}^{\top}\boldsymbol{\eta})y) \exp(\mathbf{Z}_{0,k}^{\top}\boldsymbol{\eta}).$$

## A.2 ALGORITHM DDPG FOR INTERVAL-CENSORED DATA

---

**Algorithm 1** DDPG for interval-censored Data

---

1: Randomly initialize action-value network $Q$ with parameter $\boldsymbol{\theta}^Q$.
2: Initialize target network $Q'$ with weights $\boldsymbol{\theta}^{Q'} \leftarrow \boldsymbol{\theta}^Q$.
3: Initialize replay buffer $R$, set soft update weight $\tau$ and learning rate $\alpha$.
4: **for** episode = 1 to $M$ **do**
5:     Initialize a random process $\epsilon$ for action exploration.
6:     Receive initial observation state $\{\boldsymbol{X}_{i,1}\}_{i=1}^{N}$.
7:     **for** $k = 1$ to $K$ **do**
8:         With probability $\epsilon$ select a random action $A_{i,k}$.
9:         Otherwise select $A_{i,k} = \arg \max_{a \in \mathcal{A}} Q(\boldsymbol{X}_{i,k}, a; \boldsymbol{\theta}^Q)$.
10:        Execute action $A_{i,k}$ in emulator and observe $U_{i,k}, V_{i,k}, \delta_{1,i,k}, \delta_{2,i,k}, \delta_{3,i,k}, \xi_{i,k}$ and $\boldsymbol{X}_{i,k+1}$.

11:        Store transition $\Big(\widetilde{\boldsymbol{X}}_{i,k}, \widetilde{A}_{i,k}, \widetilde{U}_{i,k}, \widetilde{V}_{i,k}, \widetilde{\delta}_{1,i,k}, \widetilde{\delta}_{2,i,k}, \widetilde{\delta}_{3,i,k}, \widetilde{\xi}_{i,k}, \widetilde{\boldsymbol{X}}_{i,k+1}\Big)$ in $R$.
12:        Sample a random mini-batch of

$$(\widetilde{\boldsymbol{X}}_{i,t}, \widetilde{A}_{i,t}, \widetilde{U}_{i,t}, \widetilde{V}_{i,t}, \widetilde{\delta}_{1,i,t}, \widetilde{\delta}_{2,i,t}, \widetilde{\delta}_{3,i,t}, \widetilde{\xi}_{i,t}, \widetilde{\boldsymbol{X}}_{i,t+1})$$

from $R$, with the batch size set to $n$.
13:        Estimate $\widehat{\Lambda}(T_k|\widetilde{\boldsymbol{X}}_{i,t}, \widetilde{A}_{i,t})$ with

$$\{\widetilde{\boldsymbol{X}}_{i,t}, \widetilde{A}_{i,t}, \widetilde{U}_{i,t}, \widetilde{V}_{i,t}, \widetilde{\delta}_{1,i,t}, \widetilde{\delta}_{2,i,t}, \widetilde{\delta}_{3,i,t}, \widetilde{\xi}_{i,t}\}_{i=1}^{n}$$

using the MLE.
14:        Calculate $G_{i,t} = -\widehat{\Lambda}(\widetilde{\alpha}_t T_t \mid \widetilde{\boldsymbol{X}}_{i,t}, \widetilde{A}_{i,t}) + \gamma \max_{A'} Q'(\widetilde{\boldsymbol{X}}_{i,t+1}, \widetilde{A}'; \boldsymbol{\theta}^{Q'})$
15:        Update $\boldsymbol{\theta}^Q$ by minimizing the loss:

$$L = \frac{1}{n} \sum_{i=1}^{n} \Big(G_{i,t} - Q(\widetilde{\boldsymbol{X}}_{i,t}, \widetilde{A}_{i,t}; \boldsymbol{\theta}^Q)\Big)^2.$$

16:        Update the target networks:

$$\boldsymbol{\theta}^{Q'} \leftarrow \tau \boldsymbol{\theta}^Q + (1 - \tau)\boldsymbol{\theta}^{Q'}.$$

17:    **end for**
18: **end for**

---

The Deep Deterministic Policy Gradient (DDPG) algorithm is a widely used actor-critic, off-policy reinforcement learning method, particularly effective in high-dimensional, continuous action spaces (Lillicrap et al., 2015). It combines policy-based and value-based approaches, providing a balanced framework for optimal policy learning. At its core, DDPG uses two networks: the actor network selects actions based on the state, while the critic network evaluates actions by estimating the action-value function. A key feature of DDPG is the replay buffer, which stores past experiences to reduce

the correlation between consecutive samples, improving learning efficiency (Mnih et al., 2015). The algorithm refines its policy over time, updating the networks with minibatches from the buffer until an optimal policy is found.

### A.3 PROOF OF THEOREM 5.5.

**Lemma A.1.** *Assume that Assumptions 5.1-5.3 hold. Then the covering number of the class* $\mathcal{L}_n = \{\log S_Y(\boldsymbol{\vartheta}_n, \mathcal{D}_n) : \boldsymbol{\vartheta}_n \in \Theta_n\}$ *satisfies*

$$N\{\epsilon, \mathcal{L}_n, L_1(P_n)\} \leq K_1 M_n^{(m+1)} \epsilon^{-(p+m+1)},$$

*for some constant* $K_1$*, where* $m = o(n^\nu)$ *with* $\nu \in (0,1)$ *is the degree of Bernstein polynomials,* $M_n = O(n^a)$ *with* $a > 0$ *controls the size of the sieve space* $\Theta_n$*, and* $p$ *is the dimension of* $\boldsymbol{\beta}$*.*

**Proof of Lemma A.1.** To investigate the covering number, first note that for any $\boldsymbol{\vartheta}_n^1 = (\boldsymbol{\beta}^1, \Lambda^1)$, $\boldsymbol{\vartheta}_n^2 = (\boldsymbol{\beta}^2, \Lambda^2) \in \Theta_n$, one can easily obtain that under Assumptions 5.1-5.3,

$$\left|\log S_Y(\boldsymbol{\vartheta}_n^1, \mathcal{D}_n) - \log S_Y(\boldsymbol{\vartheta}_n^2, \mathcal{D}_n)\right| \leq K^* \left(\left\|\boldsymbol{\beta}^1 - \boldsymbol{\beta}^2\right\| + \left\|\Lambda^1 - \Lambda^2\right\|_\infty\right),$$

for some constant $K^*$, where $\|f\|_\infty = \sup_t |f(t)|$ for a function $f$. Denote $\boldsymbol{\phi}^j = \left(\phi_0^j, \cdots, \phi_m^j\right)'$ the Bernstein coefficients corresponding to $\Lambda^j, j = 1, 2$. It is easy to show

$$\left\|\Lambda^1 - \Lambda^2\right\|_\infty = \sup_t \left|\sum_{k=0}^m \phi_k^1 B_k(t, m, \sigma, \tau) - \sum_{k=0}^m \phi_k^2 B_k(t, m, \sigma, \tau)\right|$$
$$\leq \max_{0 \leq k \leq m} \left|\phi_k^1 - \phi_k^2\right| \equiv \left\|\boldsymbol{\phi}^1 - \boldsymbol{\phi}^2\right\|_\infty.$$

Combining these results, we obtain

$$\left|\log S_Y(\boldsymbol{\vartheta}_n^1, \mathcal{D}_n) - \log S_Y(\boldsymbol{\vartheta}_n^2, \mathcal{D}_n)\right| \leq K^* \left\|\boldsymbol{\beta}^1 - \boldsymbol{\beta}^2\right\| + K^* \left\|\boldsymbol{\phi}^1 - \boldsymbol{\phi}^2\right\|_\infty.$$

It thus follows that for any $\boldsymbol{\vartheta}_n \in \Theta_n$,

$$\frac{1}{n}\sum_{i=1}^n \left|\log S_Y(\boldsymbol{\vartheta}_n^1, \mathcal{D}_n) - \log S_Y(\boldsymbol{\vartheta}_n^2, \mathcal{D}_n)\right| \leq K^* \left\|\boldsymbol{\beta} - \boldsymbol{\beta}^{(j)}\right\| + K^* \left\|\boldsymbol{\phi} - \boldsymbol{\phi}^{(j)}\right\|_\infty.$$

By Lemma 2.5 of Van de Geer (2000), one can show that $\{\boldsymbol{\beta} \in R^p, \|\boldsymbol{\beta}\|_2 \leq M\}$ is covered by $(5M/(\epsilon/(2K^*)))^p$ balls with radius $\epsilon/(2K^*)$ and $\left\{\boldsymbol{\phi} \in R^{m+1}, \sum_{0 \leq k \leq m} |\phi_k| \leq M_n\right\}$ is covered by $(5M_n/(\epsilon/(2K^*)))^{m+1}$ balls with radius $\epsilon/(2K^*)$. Therefore, the covering number of $\mathcal{L}_n$ satisfies

$$N\{\epsilon, \mathcal{L}_n, L_1(P_n)\} \leq \left(\frac{10K^*M}{\epsilon}\right)^p \cdot \left(\frac{10K^*M_n}{\epsilon}\right)^{m+1} \leq K M_n^{(m+1)} \epsilon^{-(p+m+1)}.$$

This completes the proof of Lemma A.1.

**Proof of theorem 5.5** Here we give the proof when $\pi$ is a common policy. Replace $\pi$ with $\pi^*$ in the result can get the Theorem 5.5.

*Proof.* Here we give the proof when $\pi \in \Pi$ is a common policy.

$$\hat{Q}^\pi(\mathbf{x}_t, a_t) - Q^\pi(\mathbf{x}_t, a_t)$$

$$= \mathbb{E}^{P_0, \pi}\left\{\sum_{k=t}^{\bar{K}} \gamma^{k-t}\left[\log \hat{S}_{T_k}(\alpha_k G_k | \mathbf{X}_k, A_k) - \log S_{T_k}(\alpha_k G_k | \mathbf{X}_k, A_k)\right]\middle| \mathbf{X}_t = \mathbf{x}_t, A_t = a_t\right\}$$

$$= \mathbb{E}^{P_0, \pi}\left\{\sum_{k=t}^{\bar{K}} \gamma^{k-t}\left[\Lambda_{T_k}(g_k | \mathbf{X}_k, A_k) - \hat{\Lambda}_{T_k}(g_k | \mathbf{X}_k, A_k)\right]\middle| \mathbf{X}_t = \mathbf{x}_t, A_t = a_t\right\}.$$

We can write $g_k = \alpha_k G_k$ for simplicity, and due to the definitions of $\tilde{\mathbf{X}}_k$ and $\tilde{A}_k$, the equation holds in the above expression. With the assumption, we can obtain

$$\hat{\Lambda}_{T_k}\left(g_k|\tilde{\mathbf{X}}_k, \tilde{A}_k\right) - \Lambda_{T_k}\left(g_k|\tilde{\mathbf{X}}_k, \tilde{A}_k\right) = \zeta_k(\mathcal{D}_n),$$

where $\zeta_k(\mathcal{D}_n)$ is an independent and identically distributed zero-mean process. So we have

$$
\begin{aligned}
\hat{Q}^\pi(\mathbf{x}_t, a_t) - Q^\pi(\mathbf{x}_t, a_t) &= -\sum_{k=t}^{\bar{K}} \gamma^{k-t} \mathbb{E}^{P_0, \pi}\left[\zeta_k(\mathcal{D}_n) | \mathbf{X}_t = \mathbf{x}_t, A_t = a_t\right] \\
&= -\sum_{k=t}^{\bar{K}} \gamma^{k-t} \sum_{\{\mathcal{D}_n\}} \prod_{i=1}^{N_k} \mathbb{P}(\mathbf{x}_{i,k}, a_{i,k}|\mathbf{x}_{i,t}, a_{i,t}) \zeta_k(\mathcal{D}_n),
\end{aligned}
\tag{5}
$$

where for $k \le t$, $\mathbb{P}(\mathbf{x}_{i,k}, a_{i,k}|\mathbf{x}_{i,t}, a_{i,t}) = 1$, and for $k > t$,

$$\mathbb{P}(\mathbf{x}_{i,k}, a_{i,k}|\mathbf{x}_{i,t}, a_{i,t}) = \sum_{\substack{a_{i,j}, \mathbf{x}_{i,j} \\ t+1 \le j \le k}} \pi(a_{i,t}|\mathbf{x}_{i,t}) \prod_{j=t+1}^{k} P_0(\mathbf{x}_{i,j} \mid \mathbf{x}_{i,j-1}, a_{i,j-1}) \pi(a_{i,j}|\mathbf{x}_{i,j}).$$

Note that $|\log S_Y(\boldsymbol{\vartheta}, \mathcal{D}_n)|$ is bounded under Assumptions 5.1-5.3. Without loss of generality, we assume $\sup_{\boldsymbol{\vartheta} \in \Theta} |\log S_Y(\boldsymbol{\vartheta}, \mathcal{D}_n)| \le 1$. Then $Pr[\log S_Y(\boldsymbol{\vartheta}, \mathcal{D}_n)]^2 \le Pr(\sup_{\boldsymbol{\vartheta} \in \Theta} |\log S_Y(\boldsymbol{\vartheta}, \mathcal{D}_n)|)^2 \le 1$. Let $\alpha_n = n^{-1/2+\phi_1}(\log n)^{1/2}$ with $\nu/2 < \phi_1 < 1/2$. Then $\{\alpha_n\}$ is a nonincreasing sequence of positive numbers. Also for a given $\epsilon > 0$, let $\epsilon_n = \epsilon\alpha_n$. Then for sufficiently large $n$ and any $\boldsymbol{\vartheta} \in \Theta_n$, we have

$$\text{Var}\left(P_n \log S_Y(\boldsymbol{\vartheta}, \mathcal{D}_n)\right) / (4\epsilon_n)^2 \le \frac{(1/n)Pr[\log S_Y(\boldsymbol{\vartheta}, \mathcal{D}_n)]^2}{16\epsilon^2\alpha_n^2} \le \frac{1}{16\epsilon^2 n\alpha_n^2} = \frac{1}{16\epsilon^2 n^{2\phi_1}\log n} \le \frac{1}{2}.$$

Let $P_n^0$ denote the signed measure that places mass $\pm n^{-1}$ at each of the observations $\mathcal{D}$, with the random $\pm$ signs being decided independently of the $\mathcal{D}^i$'s. Then from Pollard (1984, p. 31) and $\text{Var}\left(P_n \log S_Y(\boldsymbol{\vartheta}, \mathcal{D}_n)\right) / (4\epsilon_n)^2 \le 1/2$, the following symmetrization inequality holds

$$Pr\left(\sup_{\boldsymbol{\vartheta} \in \Theta_n} |P_n \log S_Y(\boldsymbol{\vartheta}, \mathcal{D}_n) - P \log S_Y(\boldsymbol{\vartheta}, \mathcal{D}_n)| > 8\epsilon_n\right) \le 4Pr\left(\sup_{\boldsymbol{\vartheta} \in \Theta_n} |P_n^0 \log S_Y(\boldsymbol{\vartheta}, \mathcal{D}_n)| > 2\epsilon_n\right).$$

Let $\mathfrak{D} = \{\mathcal{D}^1, \cdots, \mathcal{D}^n\}$. Given $\mathfrak{D}$, choose $\boldsymbol{\vartheta}^{(1)}, \ldots, \boldsymbol{\vartheta}^{(\kappa)}$, where $\kappa = N\{\epsilon_n/2, \mathcal{L}_n, L_1(P_n)\}$, such that

$$\min_{j \in \{1, \ldots, \kappa\}} P_n \left|\log S_Y(\boldsymbol{\vartheta}, \mathcal{D}_n) - \log S_Y\left(\boldsymbol{\vartheta}^{(j)}, \mathcal{D}_n\right)\right| < \epsilon_n/2$$

for all $\boldsymbol{\vartheta} \in \Theta_n$. For each $\boldsymbol{\vartheta} \in \Theta_n$, write $\boldsymbol{\vartheta}^*$ for the $\boldsymbol{\vartheta}^{(j)}$ at which the minimum is achieved. Note that

$$|P_n^o\left(\log S_Y(\boldsymbol{\vartheta}, \mathcal{D}_n) - \log S_Y(\boldsymbol{\vartheta}^*, \mathcal{D}_n)\right)| = \left|n^{-1}\sum_{i=1}^{n} \pm \left\{\log S_Y(\theta, \mathcal{D}^i) - \log S_Y(\theta^*, \mathcal{D}^i)\right\}\right|$$

$$\le n^{-1}\sum_{i=1}^{n} |\log S_Y(\boldsymbol{\vartheta}, \mathcal{D}_n) - \log S_Y(\boldsymbol{\vartheta}^*, \mathcal{D}_n)| = P_n |\log S_Y(\boldsymbol{\vartheta}, \mathcal{D}_n) - \log S_Y(\boldsymbol{\vartheta}^*, \mathcal{D}_n)|.$$

Then we have

$$Pr\left(\sup_{\boldsymbol{\vartheta} \in \Theta_n} |P_n^o \log S_Y(\boldsymbol{\vartheta}, \mathcal{D}_n)| > 2\epsilon_n \mid \mathfrak{D}\right)$$

$$\le Pr\left(\sup_{\boldsymbol{\vartheta} \in \Theta_n} \{|P_n^o \log S_Y(\boldsymbol{\vartheta}^*, \mathcal{D}_n)| + P_n |\log S_Y(\boldsymbol{\vartheta}, \mathcal{D}_n) - \log S_Y(\boldsymbol{\vartheta}^*, \mathcal{D}_n)|\} > 2\epsilon_n \mid \mathfrak{D}\right)$$

$$\le Pr\left(\max_j \left|P_n^o \log S_Y\left(\boldsymbol{\vartheta}^{(j)}, \mathcal{D}_n\right)\right| > 3\epsilon_n/2 \mid \mathfrak{D}\right)$$

$$\le N\{\epsilon_n/2, \mathcal{L}_n, L_1(P_n)\} \max_j P\left(\left|P_n^o \log S_Y\left(\boldsymbol{\vartheta}^{(j)}, \mathcal{D}_n\right)\right| > 3\epsilon_n/2 \mid \mathfrak{D}\right).$$

From the definition of the covering number $N\left\{\epsilon_n/2, \mathcal{L}_n, L_1\left(P_n\right)\right\}$, for each $\boldsymbol{\vartheta}^{(j)}$, there exists $\check{\boldsymbol{\vartheta}}^{(j)} \in \Theta_n$ such that $P_n\left|\log S_Y\left(\check{\boldsymbol{\vartheta}}^{(j)}, \mathcal{D}_n\right) - \log S_Y\left(\boldsymbol{\vartheta}^{(j)}, \mathcal{D}_n\right)\right| < \epsilon_n/2$. Therefore, we obtain

$$
Pr\left(\left|P_n^o \log S_Y\left(\boldsymbol{\vartheta}^{(j)}, \mathcal{D}_n\right)\right| > 3\epsilon_n/2 \mid \mathfrak{D}\right)
$$

$$
\leq Pr\left(\left\{P_n\left|\log S_Y\left(\boldsymbol{\vartheta}^{(j)}, \mathcal{D}_n\right) - \log S_Y\left(\check{\boldsymbol{\vartheta}}^{(j)}, \mathcal{D}_n\right)\right| + \left|P_n^0 \log S_Y\left(\check{\boldsymbol{\vartheta}}^{(j)}, \mathcal{D}_n\right)\right|\right\} > 3\epsilon_n/2 \mid \mathfrak{D}\right)
$$

$$
\leq Pr\left(\left|P_n^o \log S_Y\left(\check{\boldsymbol{\vartheta}}^{(j)}, \mathcal{D}_n\right)\right| > \epsilon_n \mid \mathfrak{D}\right).
$$

From Hoeffding's inequality (Pollard, 1984, Appendix B), we have

$$
Pr\left(\left|P_n^0 \log S_Y\left(\check{\boldsymbol{\vartheta}}^{(j)}, \mathcal{D}_n\right)\right| > \epsilon_n \mid \mathfrak{D}\right) = Pr\left(\left|\sum_{i=1}^n \pm \log S_Y\left(\check{\boldsymbol{\vartheta}}^{(j)}, \mathcal{D}^i\right)\right| > n\epsilon_n \mid \mathfrak{D}\right)
$$

$$
\leq 2\exp\left\{-2\left(n\epsilon_n\right)^2 / \sum_{i=1}^n \left(2 \log S_Y\left(\check{\boldsymbol{\vartheta}}^{(j)}, \mathcal{D}^i\right)\right)^2\right\}
$$

$$
\leq 2\exp\left(-n\epsilon_n^2/2\right) \quad \text{(because } \left|\log S_Y\left(\check{\boldsymbol{\vartheta}}^{(j)}, \mathcal{D}_n\right)\right| \leq 1\text{)}.
$$

Combining the inequalities above together with Lemma A.1, we obtain

$$
Pr\left(\sup_{\theta \in \Theta_n}\left|P_n^0 \log S_Y\left(\boldsymbol{\vartheta}, \mathcal{D}_n\right)\right| > 2\epsilon_n \mid \mathfrak{D}\right) \leq 2N\left\{\epsilon_n/2, \mathcal{L}_n, L_1\left(P_n\right)\right\} \exp\left(-n\epsilon_n^2/2\right)
$$

$$
\leq 2K_1 M_n^{(m+1)}\left(\epsilon_n/2\right)^{-(p+m+1)} \exp\left(-n\epsilon_n^2/2\right).
$$

Note that the right-hand side does not depend on $\mathfrak{D}$, by taking expectations over $\mathfrak{D}$, we have

$$
Pr\left(\sup_{\boldsymbol{\vartheta} \in \Theta_n}\left|P_n^0 \log S_Y\left(\boldsymbol{\vartheta}, \mathcal{D}_n\right)\right| > 2\epsilon_n\right) \leq 2K_1 M_n^{(m+1)}\left(\epsilon_n/2\right)^{-(p+m+1)} \exp\left(-n\epsilon_n^2/2\right).
$$

Also $M_n = O\left(n^a\right)$ implies that there exists a positive constant $C$ such that $0 < M_n \leq Cn^a$ for $n$ large enough, and $m = o\left(n^\nu\right)$ and $\phi_1 > \nu/2$ imply $m = o\left(n^{2\phi_1}\right)$. Combining these results with the symmetrization inequality derived above, we obtain

$$
Pr\left(\sup_{\boldsymbol{\vartheta} \in \Theta_n}\left|P_n \log S_Y\left(\boldsymbol{\vartheta}, \mathcal{D}_n\right) - P \log S_Y\left(\boldsymbol{\vartheta}, \mathcal{D}_n\right)\right| > 8\epsilon_n\right)
$$

$$
\leq 4Pr\left(\sup_{\boldsymbol{\vartheta} \in \Theta_n}\left|P_n^0 \log S_Y\left(\boldsymbol{\vartheta}, \mathcal{D}_n\right)\right| > 2\epsilon_n\right)
$$

$$
\leq 8K M_n^{(m+1)}\left(\epsilon_n/2\right)^{-(p+m+1)} \exp\left(-n\epsilon_n^2/2\right)
$$

$$
\leq 8\hat{K} \exp\left[(m+1)a \log n - (p+m+1)\left\{\log\left(\epsilon n^{-1/2+\phi_1}(\log n)^{1/2}\right) - \log 2\right\} - n\epsilon^2 n^{-1+2\phi_1} \log n/2\right]
$$

$$
\leq 8\hat{K} \exp\left[(p+m+1)\left\{(a+1/2-\phi_1)\log n - \log\log n/2 - \log\epsilon + \log 2\right\} - \epsilon^2 n^{2\phi_1} \log n/2\right]
$$

$$
\leq 8\hat{K} \exp\left(-\bar{K} n^{2\phi_1} \log n\right)
$$

where $\hat{K}$ and $\bar{K}$ are constants. Hence $\sum_{n=1}^\infty Pr\left(\sup_{\boldsymbol{\vartheta} \in \Theta_n}\left|P_n \log S_Y\left(\boldsymbol{\vartheta}, \mathcal{D}_n\right) - P \log S_Y\left(\boldsymbol{\vartheta}, \mathcal{D}_n\right)\right| > 8\epsilon_n\right) < \infty$. By the Borel-Cantelli lemma, we have $\sup_{\boldsymbol{\vartheta} \in \Theta_n}\left|P_n \log S_Y\left(\boldsymbol{\vartheta}, \mathcal{D}_n\right) - P \log S_Y\left(\boldsymbol{\vartheta}, \mathcal{D}_n\right)\right| \to 0$ almost surely. From the MDP process, we can see that the $\zeta_k(\mathcal{D}_n)$ between different k's are also independent of each other, so we can figure out that

$$
\mathbb{E}\left[\hat{Q}^\pi(\tilde{\mathbf{x}}_t, \tilde{a}_t) - Q^\pi(\mathbf{x}_t, a_t)\right] = 0,
$$

$$
Var\left[\hat{Q}^\pi(\tilde{\mathbf{x}}_t, \tilde{a}_t) - Q^\pi(\mathbf{x}_t, a_t)\right] = \sigma(P_0, \pi, \gamma, \mathbf{x}_t, a_t),
$$

where

$$\sigma(P_0, \pi, \gamma, \mathbf{x}_t, a_t) = \sum_{k=t}^{\bar{K}} \gamma^{2k-2t} \sum_{\{\mathcal{D}_n\}} w(\mathcal{D}_n)^2 \mathbb{E}[\zeta_k^2(\mathcal{D}_n)], \tag{6}$$

$$w(\mathcal{D}_n) = \prod_{i=1}^{N_k} \sum_{\substack{a_{i,j}, \mathbf{x}_{i,j} \\ t+1 \leq j \leq k}} \pi(a_{i,t}|\mathbf{x}_{i,t}) \prod_{j=t+1}^{k} P_0(\mathbf{x}_{i,j} \mid \mathbf{x}_{i,j-1}, a_{i,j-1}) \pi(a_{i,j}|\mathbf{x}_{i,j}).$$

From the form of (6), we can obtain the following expression:

$$\sigma(P_0, \pi, \gamma, \mathbf{x}_t, a_t) \leq \sum_{k=t}^{\bar{K}} \gamma^{2k-2t} \left( \sum_{\{\mathcal{D}_n\}} w(\mathcal{D}_n) \right)^2 \sup_k \mathbb{E}[\zeta_k^2(\mathcal{D}_n)] = \sum_{k=t}^{\bar{K}} \gamma^{2k-2t} \sup_k \mathbb{E}[\zeta_k^2(\mathcal{D}_n)].$$

What's more, if $\zeta_k(\mathcal{D}_n)$ follows a normal distribution, then $\hat{Q}^\pi(\tilde{\mathbf{x}}_t, \tilde{a}_t) - Q^\pi(\mathbf{x}_t, a_t)$ also follows a normal distribution according to (5) with the mean of zero and variance of $\sigma(P_0, \pi, \gamma, \mathbf{x}_t, a_t)$. □

### A.4 SUPPLEMENTAL EXPERIMENTS

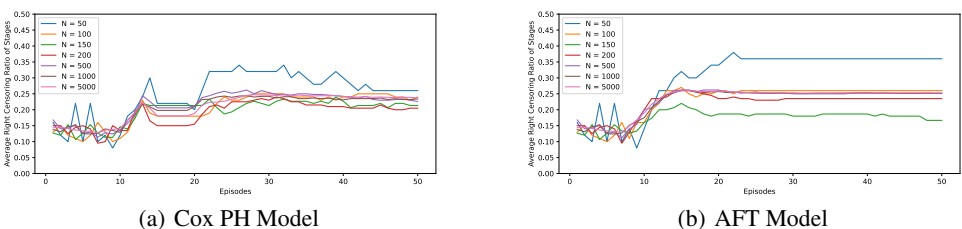

(a) Cox PH Model  (b) AFT Model

Figure 6: Right Censoring Rate of each Stage to Episodes

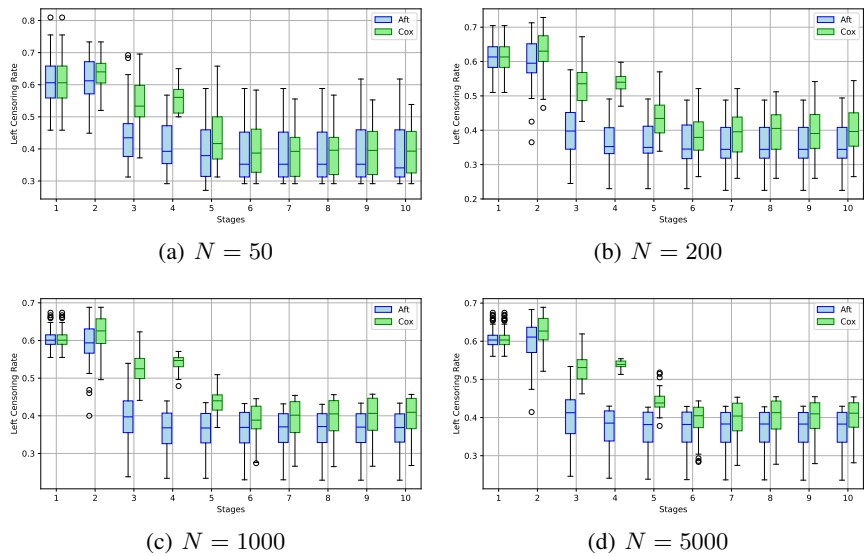

(a) $N = 50$  (b) $N = 200$

(c) $N = 1000$  (d) $N = 5000$

Figure 7: Left censoring rates of all episodes for each stage under different $N$ using Aft and Cox Model

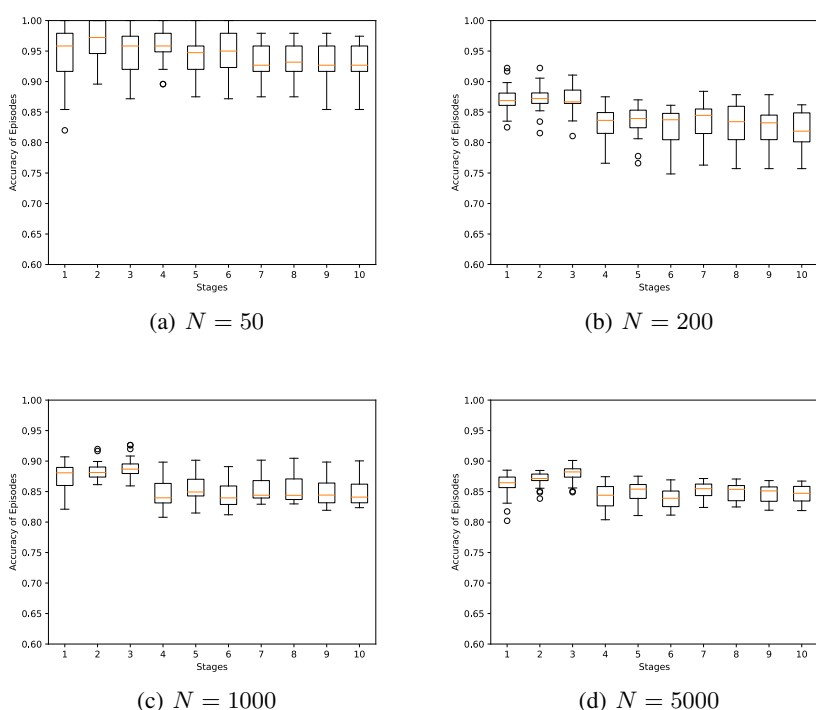

Figure 8: Comparison of Boxplot of Accuracy at each stage for different sample sizes using Cox PH Model

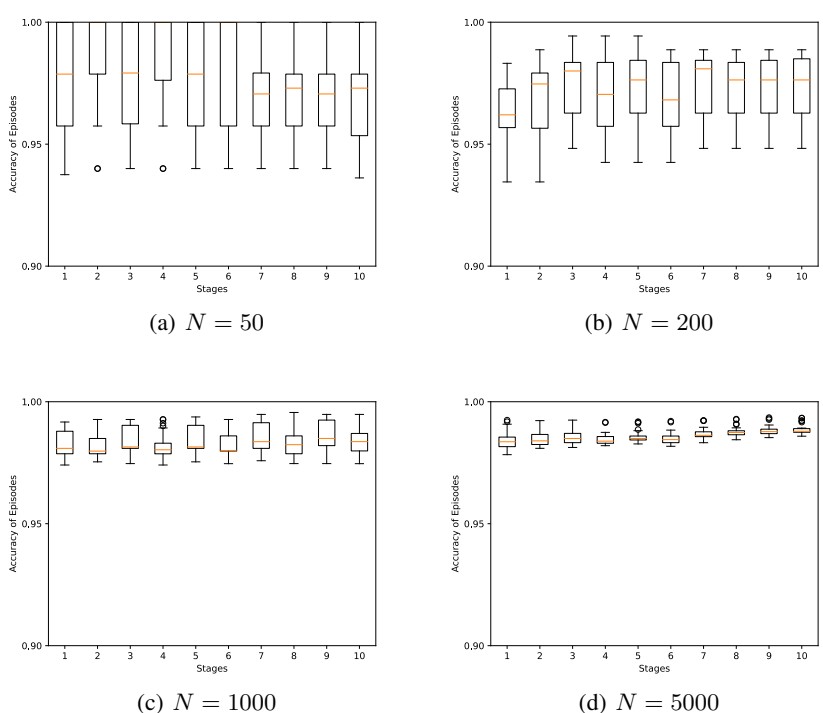

Figure 9: Comparison of Boxplot of Accuracy at each stage for different sample sizes using AFT Model

