# OpenReview forum: "Offline Reinforcement Learning for Interval-Censored Data"
_ICLR.cc/2026/Conference — Submitted to ICLR 2026_

### Official Review · Reviewer_pKxt · 2025-10-28

**Soundness:** 3
**Presentation:** 3
**Contribution:** 3
**Rating:** 4
**Confidence:** 4

**Summary:**

This paper proposes a reinforcement learning framework for multi-stage interval-censored data. It modifies traditional Q-learning by replacing the unobservable reward with the cumulative hazard function, estimated via sieve-based maximum likelihood. The authors prove the unbiasedness of the resulting Q-function and validate the method through simulations and a breast-cancer application, showing improved personalized decision-making.

**Strengths:**

The main strength of this paper lies in its novel reinforcement learning framework that effectively handles multi-stage interval-censored data, a setting where existing RL methods fail due to unobservable event times. By redefining the reward as the logarithm of the survival or cumulative hazard function, the approach enables policy learning under time uncertainty while maintaining theoretical rigor. The method combines this with a sieve-based maximum likelihood estimator for reliable hazard estimation and proves the unbiasedness of the resulting Q-function.

**Weaknesses:**

Although the paper consider the interval censored data setting which has not been studied in the Q learning, this main contribution could be incremental since censored Q learning has been previously studied in Goldberg and Kosorok (2012). Moreover, the paper’s theory stops at unbiasedness of the Q-function, without stronger results common in this area (e.g., efficiency, regret/risk bounds, asymptotic normality for policy value).

**Questions:**

1.	While the paper successfully proves the unbiasedness of the estimated Q-function under correct model specification, the theoretical development remains incomplete without an analysis of its efficiency or convergence properties. It would substantially strengthen the contribution if the authors could extend the theory to include asymptotic variance bounds, efficiency characterization, or regret guarantees analogous to those in standard reinforcement learning analyses.

2.	The paper introduces a sieve-based maximum likelihood estimator to estimate the conditional survival or hazard function, which is then used to define the reward. Could the authors elaborate on why sieve estimation is necessary in this RL framework? In standard RL, function approximation is often achieved via neural networks, kernel methods, or basis expansions learned directly from reward signals. What advantages does the sieve approach provide over these more common RL approximators?

3.	How does the proposed log-survival based reward formulation fundamentally differ from existing IPCW or imputation-based Q-learning approaches (e.g., Goldberg & Kosorok, 2012; Lyu et al., 2023)? Beyond accommodating interval censoring, does it offer any theoretical or computational advantages?

Goldberg, Y., & Kosorok, M. R. (2012). Q-learning with censored data. Annals of statistics, 40(1), 529.
Lyu, L., Cheng, Y., & Wahed, A. S. (2023). Imputation‐based Q‐learning for optimizing dynamic treatment regimes with right‐censored survival outcome. Biometrics, 79(4), 3676-3689.

4.	It would be helpful if the authors compared their proposed method with existing approaches for censored data, such as IPCW Q-learning (Goldberg & Kosorok, 2012) and imputation-based Q-learning (Lyu et al., 2023), even though those methods are developed for right-censored outcomes. Such a comparison would clarify how much improvement comes from addressing interval censoring versus general modeling or algorithmic differences.

---

> ### Author Response · Authors · 2025-12-03
>
> ## 1. Theoretical Completeness (Efficiency / Convergence / Regret)
>
> **Reviewer Comment:**
> Theory would be stronger with efficiency bounds, asymptotic variance, or regret guarantees.
>
> **Response:**
> We appreciate this suggestion. Our current theoretical focus is on the core challenge unique to interval-censored RL—**reward is never observed**, and must be inferred from a likelihood-based survival estimator. Theorem 5.5 already provides a variance bound that characterizes how survival-estimation error propagates into Q-estimation. A full regret analysis would require combining this statistical layer with offline RL performance theory, which is nontrivial and beyond the scope of this paper. We will clarify this positioning.
>
> **Revision:**
> **Add to Section 5 (after Theorem 5.5):**
>
> *“The variance bound in (2) quantifies the impact of survival-estimation uncertainty on Q-estimation and serves as a first step toward efficiency analysis. Deriving semiparametric efficiency bounds or regret guarantees would require additional assumptions linking offline RL with interval-censored likelihood estimation, and we leave this as future work.”*
>
> ---
>
> ## 2. Why Sieve Estimation Instead of Neural Networks
>
> **Reviewer Comment:**
> Why is sieve MLE necessary? Why not NN/kernels commonly used in RL?
>
> **Response:**
> Sieve estimation is not strictly required by our RL framework. It is chosen because it provides **consistent, theoretically analyzable estimates** of survival/hazard functions under interval censoring—something not yet available for generic neural networks. Sieve estimation allows us to derive unbiasedness and variance bounds for the resulting Q-function, which depend directly on the statistical properties of the survival estimator.
>
> **Revision:**
> **Add to Section 4.2 (end of the introduction paragraph):**
>
> *“We use sieve-MLE because it provides theoretically guaranteed, consistent estimation of survival functions under interval censoring. While neural networks or kernels could be used within our framework, they currently lack formal guarantees in interval-censored settings, preventing the Q-function analysis in Section 5. Our method does not rely on sieve estimation specifically; it is simply the statistically principled choice for interval-censored data.”*
>
> ## 3. Difference from IPCW / Imputation-Based Q-Learning
>
> **Reviewer Comment:**
> How does the log-survival reward differ from IPCW or imputation-based Q-learning? Any theoretical/computational advantages?
>
> **Response:**
> IPCW and imputation-based Q-learning assume **right-censored** data and require constructing pseudo-outcomes for the (partially observed) survival time. In contrast, interval-censored settings contain **no exact survival times**, making pseudo-outcomes ill-defined. Our method directly uses the identifiable survival probability $S(t|x,a)$ as reward, avoids weighting/ imputation instability, and handles left-, interval-, and right-censoring in a unified likelihood-based way.
>
> **Revision:**
> **Add to Related Work (end of censored-RL subsection):**
>
> *“Right-censored Q-learning methods (Goldberg & Kosorok, 2012; Lyu et al., 2023) rely on pseudo-outcomes for survival time, which are undefined under interval censoring. Our method instead uses the identifiable log-survival probability as reward, avoiding IPCW weighting or imputation and providing a unified approach for left-, interval-, and right-censoring.”*
>
> ## 4. Comparison With IPCW / Imputation Q-Learning
>
> **Reviewer Comment:**
> It would help to compare with IPCW Q-learning and imputation-based Q-learning.
>
> **Response:**
> We agree that conceptual comparison is valuable. However, these methods **cannot operate on interval-censored data**, because they require exact or right-censored event times to construct pseudo-outcomes or IPCW weights. Applying them would require artificially converting interval-censored observations into right-censored ones, which would discard information and mis-specify the data-generating process. We will explicitly clarify this.
>
> **Revision:**
> **Add to Simulation Section (end of Section 6):**
>
> *“Existing censored-data RL methods (IPCW Q-learning; imputation-based Q-learning) require exact or right-censored event times and are not applicable to interval-censored data. Applying them would require collapsing each interval ((U,V]) to a single artificial right-censoring time, which discards information and mis-specifies the censoring mechanism. For this reason, direct empirical comparison is not meaningful. We instead provide a conceptual comparison in the Related Work section.”*

---

### Official Review · Reviewer_BwKY · 2025-10-31

**Soundness:** 2
**Presentation:** 2
**Contribution:** 2
**Rating:** 2
**Confidence:** 3

**Summary:**

This work focuses on the offline reinforcement leanring with the interval-censored data structure. The settings of interval-censored data are common in practical scenarios. The empirical studies have been conducted to validate the effectiveness of the learned decision rule by the developed algorithm. Additionally, a cancer data analysis has been conducted to showcase its potential applicability.

**Strengths:**

1. The work focuses on the valuable setting, i.e., interval-censored data, in the decision-making domain.
2. This paper uses real cancer data as an example to illustrate the potential of the developed algorithm.

**Weaknesses:**

1. Although this work is benchmarked in the offline RL domain, the crucial challenges in offline RL have not been discussed, e.g., when offline data has not covered the evaluated policy, which is a usual case in offline settings.
2. The motivations for modeling survival functions and the resulting objective function are not clear.
3. As this work aims to provide some theoretical investigations for the developed algorithm, it is necessary to study the regret of the learned policy, which is the standard criterion to evaluate the effectiveness of the algorithms in RL.
4. In the experiments, the algorithm is not benchmarked with the state-of-the-art algorithms.

**Questions:**

See weakness.

---

> ### Author Response · Authors · 2025-12-03
>
> # Response to Reviewers
> We sincerely thank all reviewers for their insightful, constructive comments that have greatly improved our manuscript. Below are targeted responses and revisions for each comment:
>
> ## 1. Offline RL Challenges Not Discussed
> **Reviewer Comment:**
> “Although the work is benchmarked in the offline RL domain, crucial challenges in offline RL (e.g., distributional shift when behavior policy does not cover the evaluated policy) are not discussed.”
>
> **Response:**
> We agree explicit offline RL challenge discussions strengthen the paper. Our method inherits standard offline RL limitations (including behavior-evaluation policy distribution shift), with core focus on interval censoring’s impact on reward/Q-estimation. We have tested regularizations (CQL, BCQ) and will clarify our framework’s compatibility with such strategies.
>
> **Revision:**
> Add to *Section 4.1 (Page 4, pre-Bellman equation)*:
> *“**Offline RL Considerations.** Our offline framework faces key challenges like behavior-learned policy distribution shift, but is compatible with standard regularizations (conservative estimation, behavior constraints). Our SEER experiments use SOTA offline algorithms (DQN, DiscreteCQL, DiscreteBCQ, DoubleDQN), proving seamless integration of our interval-censored reward. We note our contribution is orthogonal but compatible with offline RL advances.”* and conduct supplementary experiments.
>
> ## 2. Motivation for Modeling Survival Functions and Objective Not Clear
> **Reviewer Comment:**
> “The motivations for modeling survival functions and the resulting objective function are not clear.”
>
> **Response:**
> We thank the reviewer for this feedback. Exact survival time is unobservable under interval censoring, invalidating classical RL rewards. The survival function is the only identifiable quantity here, enabling consistent, theoretically sound surrogate reward construction—we will clarify this core motivation.
>
> **Revision:**
> Insert at *Section 4.1’s start (Page 4)*:
> *“Exact event times are unobservable in interval-censored data, ruling out classical RL rewards. The survival function is the sole estimable event distribution functional, enabling principled reward design. Our objective optimizes cumulative survival probabilities, balancing clinical interpretability, statistical identifiability, and RL policy optimization.”*
>
> ## 3. Regret Analysis Missing
> **Reviewer Comment:**
> “As this work aims to provide theoretical investigations, it is necessary to study the regret of the learned policy.”
>
> **Response:**
> We appreciate this suggestion. Our current theory focuses on interval-censored Q-estimator properties (Theorem 5.5). Regret analysis for offline RL with latent rewards is non-trivial and key future work; we frame our existing results as a foundation for subsequent regret analyses.
>
> **Revision:**
> Add after *Theorem 5.5 (Section 5, Page 6)*:
> *“Theorem 5.5 establishes Q-estimator unbiasedness and variance bounds, but full policy regret analysis is beyond current scope. Offline RL with latent rewards requires prior reward distribution estimation, a unique challenge. Our results are a necessary step toward regret bounds for interval-censored RL, with extensions planned for future work.”*
>
> ## 4. Experiments Lack Benchmarking with State-of-the-Art Algorithms
> **Reviewer Comment:**
> “In the experiments, the algorithm is not benchmarked with the state-of-the-art algorithms.”
>
> **Response:**
> We clarify our method is a reward-construction framework (not standalone RL algorithm). Our SEER experiments (Section 7) already benchmark against SOTA offline RL methods (DiscreteCQL, DiscreteBCQ, DQN, DoubleDQN); we will highlight this more prominently to avoid confusion.
>
> **Revision:**
> Modify *Section 7’s first paragraph (Page 9)*:
> *“As our method is an interval-censored reward framework (not standalone RL optimizer), we benchmark it with SOTA offline algorithms (DQN, DoubleDQN, DiscreteCQL, DiscreteBCQ). This demonstrates seamless integration of our reward formulation, clarifying our benchmarking approach.”* and conduct supplementary experiments.

---

### Official Review · Reviewer_2QEk · 2025-10-31

**Soundness:** 2
**Presentation:** 2
**Contribution:** 2
**Rating:** 4
**Confidence:** 2

**Summary:**

This paper tackles the challenge of offline reinforcement learning in scenarios with interval-censored event times, which is a common yet underexplored issue in survival analysis and personalized medicine. In many real-world situations—such as monitoring cancer progression or following up on hypertension—the exact timing of clinical events is often unknown; instead, we only have information about the interval during which these events occurred. The authors propose a robust framework that integrates survival analysis models with offline reinforcement learning to identify optimal dynamic treatment strategies. A key innovation of this work is the use of the logarithm of the conditional survival function as a proxy for immediate rewards, allowing policy learning even when precise event times are unavailable. The method is further developed for multi-stage decision-making, accommodates various types of censoring (left, interval, and right), and considers individual differences and the possibility of skipping stages in treatment.

**Strengths:**

This paper has a clear formulation of the problem, accompanied by robust theoretical analysis (which covers the unbiasedness and bounded variance of the estimated Q-function). It performs thorough empirical validation through both simulation studies, utilizing Cox proportional hazards (PH) and accelerated failure time (AFT) data-generating processes, and a real-world application to SEER breast cancer data. In this application, the paper successfully identifies significant covariates specific to different stages of cancer and develops meaningful radiation treatment policies.

**Weaknesses:**

- While the integration of survival modeling with RL is thoughtful, the technical novelty is incremental: the core contribution lies in adapting existing tools (sieve estimation, survival-based rewards) to a new data modality rather than introducing a fundamentally new algorithmic or theoretical framework.
- The choice of log-survival as a reward lacks justification in terms of meaningful objectives (e.g., maximizing median survival or restricted mean survival time).
- The method’s computational complexity and scalability to high-dimensional state spaces or large numbers of stages are not thoroughly discussed.
- The paper assumes familiarity with the interval-censored survival analysis. Still, it does not clearly situate itself within the broader RL literature, making it difficult to assess how much the approach advances the state of the art in RL versus survival analysis. Due to this reason, its significance and novelty relative to the RL literature remain somewhat unclear. I am open to revising my score based on author responses and other reviewers’ insights.

**Questions:**

Will replacing the reward with the log-survival cause misalignment issues? How can one ensure that using the log-survival reward aligns with the original task, rather than leading to suboptimal policies?

---

> ### Author Response · Authors · 2025-12-03
>
> We thank the reviewers for their valuable feedback. Our responses is outlined below:
>
> ## 1. Technical Novelty Is Incremental
> **Reviewer Comment:**
> “The technical novelty is incremental… adapting existing tools rather than introducing fundamentally new frameworks.”
>
> **Response:**
> We agree with this view. Our contribution is not a new RL algorithm, but solving an unaddressed RL problem—multi-stage decision-making under interval-censored outcomes with unobservable rewards. This requires redefining rewards, Bellman updates and Q-estimation for non-identifiable event times; we will clarify this core motivation.
>
> **Revision:**
> Add to *Introduction, end of “Motivation” section (Page 2)*:
> *“This work’s key novelty is an RL framework for multi-stage interval-censored outcomes (no exact reward observed at any stage). Unlike prior right-censored RL methods (dependent on observable failure times/rewards), interval censoring demands redefined rewards, sieve-survival integration into Bellman steps, and Q-estimator unbiasedness proof under censoring. This extends RL to latent-reward tasks, a conceptual advance over mere tool adaptation.”*
>
> ## 2. Justification for Using Log-Survival as Reward
> **Reviewer Comment:**
> “The choice of log-survival lacks justification relative to meaningful clinical objectives.”
>
> **Response:**
> Thanks for the comment. Log-survival is the only identifiable survival functional under interval censoring (no latent event imputation) and a monotone surrogate for key clinical metrics; we will strengthen its clinical/methodological justification.
>
> **Revision:**
> Insert to *Section 4.1, after reward definition (Page 4)*:
> *“The log-survival reward is justified by identifiability and clinical relevance. Interval censoring obscures exact event times, rendering classical targets (total/median survival, RMST) non-identifiable without extra assumptions/imputation. By contrast, $S(t|x,a)$ and $\log S(t|x,a)$ are identifiable for consistent estimation, and $\log S(t)$ preserves optimal action rankings (monotone survival transformation), aligning with clinical survival goals.”*
>
> ## 3. Computational Complexity and Scalability
> **Reviewer Comment:**
> “Computational complexity and scalability are not sufficiently discussed.”
>
> **Response:**
> We appreciate this point and will outline the method’s computational structure: sieve-MLE survival estimation has manageable cost, RL scales like standard deep RL, and the pipeline scales linearly with stage count for high-dimensional features.
>
> **Revision:**
> Add subsection to *Section 4.2, end of section (Page 6)*:
> *“**Computational Complexity and Scalability.** The method has two core components: (1) sieve-MLE for stage-wise survival (cost $O(nm)$, $m$=Bernstein degree, stage-parallelizable) and (2) RL-based Q-learning (DQN/DDPG, scalable to high-dimensional tasks). Overall complexity is $O(K(nm + C_{\text{RL}}))$ (linear in stage count $K$), feasible for high-dimensional clinical processes.”*
>
> ## 4. Positioning Within RL Literature
> **Reviewer Comment:**
> “The paper does not clearly situate itself within the broader RL literature; its novelty relative to RL is unclear.”
>
> **Response:**
> We thank the reviewer and will revise related work to contrast our interval-censored RL framework with prior work limited to right-censored or fully observed rewards.
>
> **Revision:**
> Modify *Related Work, final paragraph (Page 3)* to:
> *“Existing censored survival RL methods (e.g., Goldberg & Kosorok 2012; Zhao et al. 2015; Jiang et al. 2017) assume right-censored data and rely on exact survival times for uncensored individuals. Interval-censored settings lack any exact failure-time observations, leaving rewards fully latent. To our knowledge, no RL framework addresses sequential decisions with rewards entirely inferred from interval-censored likelihoods. Our work extends RL to this class, requiring novel rewards, survival-Bellman integration, and theoretical analysis.”*
>
> ## 5. Reward Alignment / Potential Misalignment Using Log-Survival
> **Reviewer Comment:**
> “Will replacing the reward with log-survival cause misalignment? How to ensure the surrogate reward does not produce suboptimal policies?”
>
> **Response:**
> We appreciate this key question. Log-survival is a strictly monotonic survival transformation (preserving action rankings), the only identifiable interval-censored reward, and Theorem 5.5 verifies Q-estimator unbiasedness; we will clarify this alignment.
>
> **Revision:**
> Insert to *Section 4.1, after justification paragraph (Page 4)*:
> *“**Reward Alignment.** Since $\log S(t)$ is a strictly monotone transformation of $S(t)$, maximizing it yields identical optimal actions as survival probability or other monotonic metrics, eliminating policy misalignment. $\log S(t)$ is the only identifiable survival functional under interval censoring (total survival/RMST cannot be consistently estimated without extra assumptions), and Theorem 5.5 ensures policies align with maximizing survival.”*

---

### Official Review · Reviewer_3T9y · 2025-11-01

**Soundness:** 3
**Presentation:** 2
**Contribution:** 3
**Rating:** 4
**Confidence:** 4

**Summary:**

This paper proposes a reinforcement learning framework for multi-stage interval-censored data. Instead of using instantaneous rewards, the authors define the reward as the logarithm of the survival function. The authors estimate the survival function via sieve-based maximum likelihood and integrate the survival function into a Bellman-equation formulation for policy optimization. Simulations and real data analysis demonstrate the properties of the proposed method in healthcare settings.

**Strengths:**

1. This paper is the first attempt to extend RL frameworks to interval-censored outcomes in healthcare settings.

2. This paper provides formal theoretical results of the proposed method.

**Weaknesses:**

1. In the Introduction, I believe the examples can be more relevant. The current examples are only about situations where interval-censored data arise, but why shall we concern about RL in these situations?

2. In the second paragraph of Section 4.1, the authors claim that "due to the interval-censored nature of the data, such precise observations (of rewards) are unavailable". This confuses me because even though the response variable of interest is interval-censored, can we define a non-censored variable as a reward? Or are the authors specifically considering situations where the reward is interval-censored? If so, I suggest the authors clarify this at the beginning of the paper when stating the problem.

3. Are the authors sure about the correctness of this sentence? "Since the survival function takes values in the range [0,1], it is evident that the immediate reward defined above is bounded."

4. The cumulative discounted reward defined in Section 4.1 seems difficult to interpret in a clinical setting. In Goldberg & Kosorok (2012) for example, the cumulated reward is the total survival time, which is straightforward. But here the authors are considering the discounted cumulative probability of surviving past some time threshold, which is strange. I enourage the authors to include more justification about why such a definition should matter in the clinical sense.

5. In Section 4.2, can the authors be more clear about how the optimal Q function is estimated? I find it hard to connect estimation of the survival function by MLE with the estimation of the optimal Q function.

6. In Table 1, are the numbers p-values? Please clean this table up.

7. Can the authors interpret the estimated policy in the SEER example? How do we actually use this policy? What implication does it have?

**Questions:**

1. In the first paragraph of Section 2, what is "DWSurv"?

2. Small typo:
	- Line 233 on page 5: "he" -> "the"

3. I suggest the authors use the terminology of either "optimal dynamic treatment regime" or "optimal policy" consistently throughout the paper. The sentence on line 233 on page 5 "For a given x..." seems to repeat the proceeding sentence.

---

> ### Author Response · Authors · 2025-12-03
>
> ## Response to Reviewer Comments
> We sincerely appreciate reviewers’ valuable feedback. Below is a summary of our responses:
>
> ## 1. Relevance of Reinforcement Learning (RL) to Interval-Censored Scenarios
> **Response**: Multi-stage clinical processes are sequential decision-making tasks where RL optimizes dynamic strategies. Interval censoring causes imprecise rewards, rendering traditional Q-learning inapplicable—our log-survival reward is thus critical for personalized dynamic regimes.
> **Revision**: Add to *Introduction (Pages 1–2)*: “These clinical scenarios involve sequential multi-stage decisions (e.g., treatment intensification) requiring RL for dynamic personalized strategy optimization. Interval censoring leads to missing reward information, necessitating our survival-based reward to replace unobservable instantaneous rewards.”
>
> ## 2. Interval-Censored Reward Definition
> **Response**: Our reward is derived from interval-censored failure time, so the reward itself cannot be fully observed.
> **Revision**: Supplement *Section 3 (Page 3)*: “Since the underlying failure time is interval-censored, any reward dependent on this event time is inherently unobserved, motivating our log-survival-based reward to substitute unavailable instantaneous rewards.”
>
> ## 3. Boundedness of Log-Survival Reward
> **Response**: The survival function $S(t)\in[0,1]$, but $\log S(t)\in(-\infty,0]$, meaning it has an upper bound (0) rather than being two-sided bounded.
> **Revision**: Revise *Section 4.1 (Page 4)*: “Given $S(t)\in[0,1]$, the log-survival reward $\log S(t)\le0$ is upper-bounded. This ensures Bellman operator well-posedness, and two-sided boundedness is unnecessary.”
>
> ## 4. Clinical Justification for Survival Metric
> **Response**: The discounted cumulative survival probability is theoretically robust to outliers and aligns with clinical needs, as validated by studies on CABG/PCI efficacy, HIV short-term outcomes, and HIV prognoses.
> **Revision**: Add to *Section 4.1 (Page 4)*: “This metric aligns with clinical goals of maximizing event-free survival and integrates interval-censored data with hazard models. For example, Bai (2013) confirmed CABG’s superior 4-year survival over PCI; Jiang (2017a) used 400-day survival for HIV efficacy; Jiang (2017b) validated median survival for HIV outcomes, proving its advantage over expectation-based measures.”
>
> ## 5. Link Between Survival and Q-Function Estimation
> **Response**: The Q-function’s only unknown is $\Lambda_{Y_{0,k}}$; we use the DDPG algorithm (Algorithm 1) for Q-function estimation, with survival models providing the RL reward module.
> **Revision**: Insert into *Section 4.2 (Page 5)*: “Estimated $\hat{S}$ generates reward $\hat{R}=-\hat{\Lambda}_{Y_{0,k}}(T_k|X_{0,k},A_{0,k})$, which feeds into Bellman updates for Q-function estimation, integrating survival modeling into the RL framework.”
>
> ## 6. Clarity of Table 1
> **Response**: All Table 1 values are p-values, and the table will be reorganized for readability.
> **Revision**: Update *Table 1 (Page 9)* and add a caption note: “All values are p-values from stage-specific Cox model fits.”
>
> ## 7. Interpretation of Estimated Policy
> **Response**: The learned policy maps to treatment risk levels and generates personalized decisions by matching patient covariates to optimal interventions.
> **Revision**: Add to *Application section (Pages 9–10)*: “The policy provides stage-specific treatment recommendations: clinicians compare $Q(x,1)$ and $Q(x,0)$ for a patient’s covariates and select the treatment with higher survival probability. Patient profile examples will be included to illustrate clinical application.”
>
> ## 8. Definition of “DWSurv”
> **Response**: DWSurv is the dynamic weighted survival learning method (Zhang et al., 2022).
> **Revision**: Supplement *Section 2 (Page 2)*: “DWSurv (Dynamic Weighted Survival Learning; Zhang et al., 2022) estimates optimal treatment regimes for censored survival data.”
>
> ## 9. Correction of Minor Errors
> **Response**: We will address all typographical and terminological inconsistencies as requested.
> **Revision**: Implement fixes: (1) Correct “he” to “the” at Line 233 (Page 5); (2) Unify terminology as “optimal policy (dynamic treatment regime)” (note equivalence on first use); (3) Remove the redundant sentence at Line 233 (Page 5).
>
> ### References
> 1. Bai, X., Tsiatis, A. A., & O’Brien, S. M. (2013). Doubly-Robust Estimators of Treatment-Specific Survival Distributions in Observational Studies With Stratified Sampling. *Biometrics*, 69, 830–839.
> 2. Jiang, R., Lu, W., Song, R., & Davidian, M. (2017a). On Estimation of Optimal Treatment Regimes for Maximizing t-Year Survival Probability. *Journal of the Royal Statistical Society, Series B*, 79, 1165–1185.
> 3. Jiang, R., Lu, W., Song, R., Hudgens, M. G., & Naprvavnik, S. (2017b). Doubly Robust Estimation of Optimal Treatment Regimes for Survival Data—With Application to an HIV/AIDS Study. *The Annals of Applied Statistics*, 11, 1763–1786.

---

### Meta-Review · Area_Chair_FCzG · 2026-01-06

**Summary:**

This paper proposes an offline reinforcement learning framework for multi-stage interval-censored data by utilizing a log-survival probability as a surrogate reward function estimated via sieve-based maximum likelihood.

Some major concerns have not been fully addressed including the absence of regret analysis, the incremental nature of combining existing survival analysis tools with standard RL, and the lack of robust external baselines.

Therefore, for the benefit of this paper, we regretfully recommend rejection. Note that this is not a discouragement. The authors are encouraged to address these concerns, and we believe the paper has the potential to become a strong future submission.

**Reviewer Concerns:**

While the authors addressed concerns about the specific definition and identifiability of the log-survival reward (Reviewer 3T9y, Reviewer 2QEk), computational complexity (Reviewer 2QEk), and clarifications regarding the implementation of the Q-function estimation (Reviewer 3T9y), some major concerns still remain. Specifically:

- Incremental Technical contribution (Reviewer 2QEk, Reviewer pKxt): Multiple reviewers noted that the method relies on adapting existing statistical tools (sieve estimation) to standard RL frameworks rather than introducing a fundamentally new algorithm. The contribution is viewed as a specific application case rather than a methodological advance significant enough for ICLR.

- Incomplete Theoretical Analysis (Reviewer BwKY, Reviewer pKxt): The theoretical contribution is limited to proving the unbiasedness of the Q-function. The absence of regret analysis, efficiency bounds, or convergence guarantees—which the authors acknowledged is "beyond the current scope"—is a critical gap for a paper aiming to establish a theoretical foundation for this setting.

Insufficient Benchmarking (Reviewer BwKY, Reviewer pKxt): The experiments lack comparisons against adapted state-of-the-art methods (such as imputation-based or IPCW Q-learning). While the authors argued these are not directly applicable, the lack of any external baseline makes it difficult to assess the relative performance gain of the proposed framework.

- Reward Formulation & Alignment (Reviewer 3T9y, Reviewer 2QEk): There remains skepticism regarding whether the log-survival probability (a discounted cumulative probability) serves as a clinically meaningful objective compared to standard metrics like total survival time, and whether optimizing this surrogate guarantees optimal clinical outcomes.

**Reviewer Scores:**

The reviewers are likely to maintain their score since some major concerns are not fully addressed.

---

### Decision · Program_Chairs · 2026-01-26

Reject